# Lightweight MSA Design Advances Protein Folding From Evolutionary Embeddings

## Abstract

Protein structure prediction often hinges on multiple sequence alignments (MSAs), which underperform on low-homology and orphan proteins. We introduce PLAME, a lightweight MSA design framework that leverages evolutionary embeddings from pretrained protein language models to generate MSAs that better support downstream folding. PLAME couples these embeddings with a conservation–diversity loss that balances agreement on conserved positions with coverage of plausible sequence variation. Beyond generation, we develop (i) an MSA selection strategy to filter high-quality candidates and (ii) a sequence-quality metric that is complementary to depth-based measures and predictive of folding gains. On AlphaFold2 low-homology/orphan benchmarks, PLAME delivers state-of-the-art improvements in structure accuracy (e.g., lDDT/TM-score), with consistent gains when paired with AlphaFold3. Ablations isolate the benefits of the selection strategy, and case studies elucidate how MSA characteristics shape AlphaFold confidence and error modes. Finally, we show PLAME functions as a lightweight adapter, enabling ESMFold to approach AlphaFold2-level accuracy while retaining ESMFold-like inference speed. PLAME thus provides a practical path to high-quality folding for proteins lacking strong evolutionary neighbors.

## 1 Introduction

Understanding complex and dynamic protein structures is fundamental to target identification, validation, and drug-target interaction studies in drug design (Baker & Sali, 2001; Khoury et al., 2014). Recent advances such as AlphaFold have revolutionized structural biology, achieving near-experimental accuracy across a broad spectrum of proteins and complexes (Jumper et al., 2021; Ahdritz et al., 2024a; Abramson et al., 2024). However, most state-of-the-art folding pipelines heavily rely on evolutionary information encoded within multiple sequence alignments (MSAs) (Lin et al., 2023; Abramson et al., 2024). Consequently, their accuracy is highly correlated with the quality and depth of available MSAs. This dependency creates failure modes in low-homology families and orphan proteins (those lacking or having few evolutionary neighbors) (Kwon et al., 2021; Webb & Sali, 2016), where even small amounts of noisy or misaligned sequences can dominate the signal.

Historically, two primary classes of techniques have been developed to address weak homology. Physics-based modeling searches for low-energy conformations in energy space through handcrafted or learned force fields, but is often

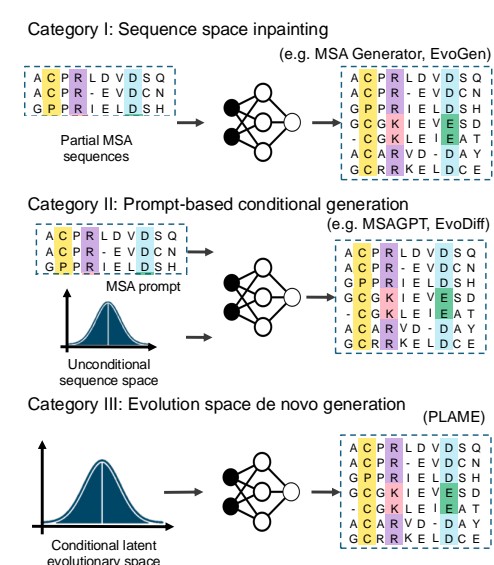

Figure 1: Taxonomy of MSA designers. Most prior work models MSAs through sequence inpainting or prompt-based generation, while PLAME directly generates MSAs *de novo* in evolutionary embedding space without prompts.

computationally intensive and limited by approximations in the energy landscape (Rohl et al., 2004; Cornell et al., 1995). Template-based methods leverage homology detection and profile-profile alignment to transfer structural priors from known folds to novel sequences (Hildebrand et al., 2009; Finn et al., 2011), but suffer degraded performance in the absence of evolutionary signals, making them unsuitable for orphan proteins. These limitations have motivated a shift toward data-driven strategies that focus on *improving the MSA itself* rather than solely the downstream folding networks.

Recent MSA design approaches can be broadly categorized into two paradigms (Figure 1). *Sequence-space inpainting* methods (e.g., MSA Generator, EvoGen) directly learn patterns in discrete sequence space to augment partial alignments, aiming to reconstruct evolutionary constraints from existing MSAs (Zhang et al., 2023; 2022). *Prompt-based conditional generation* approaches (e.g., MSAGPT, EvoDiff) utilize pre-trained models to synthesize additional sequences under MSA-style prompts (Chen et al., 2024; Alamdari et al., 2023). These methods can deepen alignments and improve folding accuracy when homologous sequences exist. An orthogonal line of research bypasses explicit MSA construction by building *implicit* evolutionary representations from single sequences through large protein language models (PLMs), as demonstrated by ESMFold (Lin et al., 2023). While MSA-free models avoid the homology bottleneck, they also forgo explicit template usage and enhanced homology signals, which may limit ultimate folding accuracy in challenging scenarios.

Despite existing progress, two critical gaps remain in structure prediction for low-homology proteins. **(i) Supervision bias:** Methods trained on existing MSA databases inherit biases toward well-studied families, limiting effectiveness for low-homology and orphan proteins. **(ii) Weak alignment-folding correlation:** Current approaches lack lightweight metrics linking MSA characteristics to folding outcomes. Sequence-based generative objectives may not align with factors that improve structural accuracy, while existing solutions like fine-tuning folding models (Chen et al., 2024) are computationally expensive and lack universal applicability.

In this study, we propose **PLAME**, motivated by the critical need to enhance structure prediction for low-homology proteins where traditional MSA-based approaches fail due to insufficient evolutionary signals. Our approach makes the following key contributions:

1. **Embedding-space MSA generation with conservation-diversity optimization:** Inspired by PLMs' success in MSA-related tasks (Hong et al., 2024; Wang et al., 2024; McWhite et al., 2023), we develop the first MSA designer that *generates auto-regressively within the evolutionary embedding space of pre-trained PLMs* rather than discrete sequences (Fig2). We further propose a novel conservation-diversity loss that captures conserved regions while extracting diverse variants from ESM embeddings with theoretical guarantee (AppendixA). The lightweight design enables PLAME to synthesize evolutionary neighborhoods even with scarce homologous sequences, achieving up to three orders of magnitude speedup while maintaining template compatibility (Table 4).

2. **HiFiAD: A principled MSA quality assessment framework:** To address the current weak alignment-folding correlation problem, we propose **H**igh-**Fi**delity **A**ppropriate **D**iversity (**HiFiAD**), a lightweight algorithm for MSA filtering that simultaneously considers site-wise conservation and inter-MSA diversity. This provides the first model-agnostic, computationally efficient criterion for selecting high-quality alignments that directly correlate with improved folding outcomes.

3. **Comprehensive validation across challenging scenarios:** On challenging low-homology and orphan datasets, PLAME consistently improves folding accuracy in both AlphaFold2 and AlphaFold3, performing similarly to DHR (Hong et al., 2024), AI-based MSA searching approach. In ablation studies, HiFiAD demonstrates performance gains across all baselines (Table1). Moreover, case studies on general and *de novo* proteins further demonstrate PLAME's generalizability while providing novel perspectives on structure enhancement from an MSA design standpoint (Table8). PLAME offers new insights and possibilities for folding enhancement through principled MSA optimization.

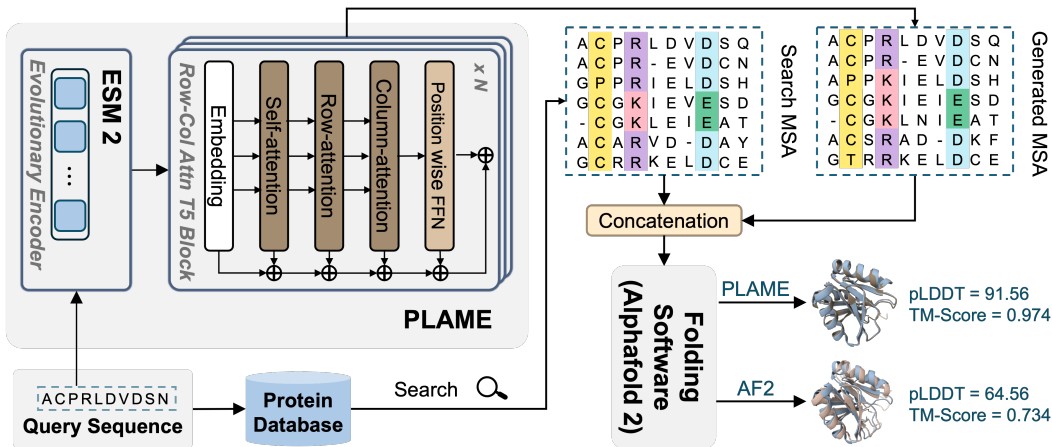

Figure 2: Overview of PLAME framework. PLAME captures ESM-2 evolutionary representations, generating MSAs for augmenting the original MSAs. The augmented MSAs serve as the homology template for folding softwares for folding enhancement. In each block of the T5-architecture, additional row-attention and col-attention are applied to capture co-evolutionary information.

## 2 METHOD

### 2.1 PROBLEM FORMULATION

Protein structure prediction relies heavily on high-quality MSAs to provide evolutionary information, but the accuracy of folding software $\mathcal{F}_\omega$ significantly drops when MSAs are sparse or insufficient. Given proteins $\mathbf{P} = \{\mathbf{s}, \mathbf{x}, \mathbf{M}\}$, where $\mathbf{s} \in \mathcal{S}$ are query sequences, $\mathbf{x} \in \mathcal{X}$ are 3D structures, and $\mathbf{M} = \{m_1, m_2, \ldots, m_n\} \in \mathcal{M}$ are MSAs with each $m_i$ as an aligned homologous sequence. The goal of MSA design models $\mathbf{p}_\theta : \mathcal{M} \to \mathcal{M}$ is designing augmented MSAs $\mathbf{M}_{\text{aug}}$ that enhances evolutionary information to obtain more accurate structures $\mathbf{x}'$ using folding software $\mathcal{F}_\omega$.

$$\mathbf{M}' = \mathbf{p}_\theta(\mathbf{M}), \quad \mathbf{x}' = \mathcal{F}_\omega(\mathbf{s}, \mathbf{M}_{\text{aug}}) \tag{1}$$

where the augmented MSAs are composed of original MSAs $\mathbf{M}$ and generated MSAs $\mathbf{M}'$, denoted as $\mathbf{M}_{\text{aug}} = \{\mathbf{M}, \mathbf{M}'\}$. The quality of the enhanced structures is evaluated using several metrics, including RMSD, TM-score, and pLDDT (See details in Section3). The key to high-fidelity MSA generation lies in constructing an informative evolutionary distribution $\mathbf{z}_{\text{evo}}$, which serves as the foundation for generating augmented MSAs $\mathbf{M}_{\text{aug}}$. Current methods utilize deep neural networks $\mathbf{f}_\theta$ to learn hidden evolutionary distributions directly from existing MSAs.

$$\mathbf{z}_{\text{evo}} = \mathbf{f}_\theta(\mathbf{M}) \tag{2}$$

However, relying solely on sequence-level information from MSAs fails to capture the complete evolutionary landscape, particularly when MSA coverage is sparse or incomplete. To overcome this limitation, we propose an evolutionary space based on evolutionary embeddings derived from pretrained protein language models (PLMs) $\mathbf{g}_\phi$.

$$\mathbf{z}_{\text{evo}} = \mathbf{f}_\theta(\mathbf{g}_\phi(\mathbf{s})) \tag{3}$$

### 2.2 MODEL ARCHITECTURE

**PLAME** employs an encoder-decoder transformer architecture similar to MSA Transformer (Rao et al., 2021), with adjustments to the T5 block structure (Vaswani, 2017). The encoder and decoder incorporate additional row-wise and column-wise attention mechanisms to better capture evolutionary patterns in MSA data (detailed in Fig 2), which is similarly applied in MSAGenerator (Zhang et al., 2023) and MSAGPT (Chen et al., 2024). Additional mechanisms are introduced as follows.

**Row Attention** Row attention models inter-sequence dependencies by summarizing evolutionary relationships across MSA depth. Given input $\mathbf{H}_{\text{enc}} \in \mathbb{R}^{B \times N \times L \times d}$, where $B$ is batch size, $N$ is MSA

depth, $L$ is sequence length, and $d$ is hidden dimension, we compute a global representation by depth-wise averaging:

$$\mathbf{H}_r = \frac{1}{N} \sum_{i=1}^{N} \mathbf{H}_{\text{enc}}[i, :, :] \in \mathbb{R}^{B \times L \times d}, \tag{4}$$

where $\mathbf{H}_r$ encodes the evolutionary space for cross-attention during decoding:

$$\text{Row-Attn}(\mathbf{Q}_r, \mathbf{K}_r, \mathbf{V}_r) = \text{softmax}\left(\frac{\mathbf{Q}_r \mathbf{K}_r^\top}{\sqrt{h}}\right) \mathbf{V}_r. \tag{5}$$

**Column Attention**   Column attention captures positional conservation patterns across MSA columns. We permute the decoder input $\mathbf{X}_{\text{dec}} \in \mathbb{R}^{B \times N \times L \times d}$ to $\tilde{\mathbf{X}}_{\text{dec}} \in \mathbb{R}^{B \times L \times N \times d}$ (swapping MSA depth $N$ and length $L$ axes) and compute cross-column attention with:

$$\mathbf{Q}_c = \tilde{\mathbf{X}}_{\text{dec}} \mathbf{W}_q, \quad \mathbf{K}_c = \tilde{\mathbf{H}}_{\text{enc}} \mathbf{W}_k, \quad \mathbf{V}_c = \tilde{\mathbf{H}}_{\text{enc}} \mathbf{W}_v$$

$$\text{Col-Att}(\mathbf{Q}_c, \mathbf{K}_c, \mathbf{V}_c) = \left( \text{softmax}\left(\frac{\mathbf{Q}_c \mathbf{K}_c^\top}{\sqrt{h}}\right) \mathbf{V}_c \right)^\top. \tag{6}$$

**Generation & Inference**   ESM2 (Lin et al., 2023) encodes the query sequence $\mathbf{s}$ into evolutionary embeddings $\mathbf{H}_{\text{input}}$. The encoder processes these through $N$ modified T5 layers:

$$\mathbf{H}_{\text{Enc}}^{(l)} = \mathbf{Enc}^{(l)}(\mathbf{H}^{(l-1)}), \quad l = 1, \dots, N, \quad \mathbf{H}^{(0)} = \mathbf{H}_r. \tag{7}$$

The decoder autoregressively generates tokens conditioned on encoder output and previous tokens:

$$\mathbf{y}_t = \mathbf{Dec}(\mathbf{y}_{<t}, \mathbf{H}_{\text{Enc}}^{(N)}). \tag{8}$$

Output embeddings are passed through softmax to produce token probabilities.

## 2.3   CONSERVATION-DIVERSITY TRAINING LOSS

We propose a position-aware causal inference approach for diverse MSA generation, integrating a **PSSM-Weighted Cross-Entropy** (**PCE**) Loss and a **DIversity REgularization** (**DIRE**) Loss to balance focus on conserved regions with sampling diversity.

**PCE Loss**   The PCE Loss emphasizes accurate predictions in conserved regions of the MSA, which are critical for maintaining protein structure and function. For a single sequence, it is defined as:

$$\mathcal{L}_{\text{seq}} = -\sum_{l=1}^{L} w_l \cdot \log p(y_l \mid y_{<l}), \tag{9}$$

where $L$ denotes sequence length, $y_l$ denotes the amino acid at site $l$, and $p(y_l \mid y_{<l})$ denotes the predicted discrete probability distribution of $y_l$. The position-specific weights $w_l$ are derived from the Position-Specific Scoring Matrix (PSSM) (Henikoff & Henikoff, 1994) and reflect the conservation level at each position. These weights are normalized to the range $[1 - \delta, 1 + \delta]$, where $\delta$ controls sensitivity to conservation. Specifically,

$$w_l = 1 + \delta \cdot \frac{\text{freq}_l - \min(\text{freq})}{\max(\text{freq}) - \min(\text{freq})}. \tag{10}$$

where freq denotes the residue-frequency of 20 types of amino acids. During model training, we apply $\delta = 0.5$, assigning higher weights to conserved positions and lower weights to less conserved ones. For a batch of $N$ sequences, the PCE loss averages over all sequences and positions:

$$\mathcal{L}_{\text{PCE}} = -\frac{1}{N} \sum_{j=1}^{N} \sum_{l=1}^{L_j} w_l^{(j)} \cdot \log p(y_l^{(j)} \mid y_{<l}^{(j)}), \tag{11}$$

where $L_j$ is the length of the $j$-th sequence, and $w_l^{(j)}$ is the weight for position $l$ in sequence $j$. This loss emphasizes conserved regions while allowing flexibility in less conserved areas.

**DICE Loss**  The DIRE loss promotes sequence diversity by maximizing amino acid entropy:

$$\mathcal{L}_{\text{DIRE}} = -\frac{1}{N} \sum_{j=1}^{N} \frac{1}{L_j} \sum_{l=1}^{L_j} H_l^{(j)}, \tag{12}$$

where $H_l^{(j)} = -\sum_{a \in \mathcal{A}} p(a \mid y_{<l}) \log p(a \mid y_{<l})$ is the entropy at position $l$ in sequence $j$, and $\mathcal{A}$ is the set of all amino acids.

**Combined Loss Function**  The combined loss function balances conservation and diversity:

$$\mathcal{L} = \alpha \cdot \mathcal{L}_{\text{PCE}} + (1 - \alpha) \cdot \mathcal{L}_{\text{DIRE}}, \tag{13}$$

with $\alpha = 0.9$ prioritizing conservation while maintaining variability. Our theoretical analysis in AppendixA demonstrates that PCE Loss enhances the model's understanding of MSA profile, while DIRE Loss functions as a regularizer to prevent neglect of variable regions.

## 2.4 MSA SELECTION METHOD – HiFiAD

Starting from MSAGPT's (Chen et al., 2024) systematic study of selection strategies showing that naive similarity-based or trimming methods can hurt performance while diversity and structure-aware filtering help but often require expensive AF2 calls. Building on this, we designed HiFiAD as a lightweight, model-agnostic selection rule that combines BLOSUM-based fidelity with recovery-based diversity to avoid both over-conserved and overly noisy sequences.

HiFiAD addresses two key challenges: (i) over-conserved sequences that distort evolutionary distributions when over-concatenated, and (ii) lack of systematic quality assessment for generated MSAs, by combining sequence similarity (fidelity) with diversity to maintain balanced evolutionary signals. Given a query sequence $s$ and generated MSAs $M = \{m_1, m_2, \ldots, m_n\}$, we define:

$$S_{\text{BLOSUM}}(m_i, s) = \sum_{j=1}^{L} B(s_j, m_{ij}), \quad \forall m_i \in M, \tag{14}$$

$$R(m_i, s) = \frac{1}{L} \sum_{j=1}^{L} \mathbb{I}[s_j = m_{ij}], \quad \forall m_i \in M, \tag{15}$$

where $B$ is the BLOSUM62 matrix, $R(m_i, s)$ is the recovery rate, and $\mathbb{I}[\cdot]$ is the indicator function. **Zero-shot selection** (Orphan proteins): Select top-$k$ sequences by $S_{\text{BLOSUM}}$ and sequences from top/bottom $k/2$ of recovery rate distribution, similar to the Static Diversity Strategy of MSAGPT. **Few-shot selection** (Low homology proteins): Limit augmented MSAs to $N_{\max} = \max(16, 2N_{\text{orig}})$ where $N_{\text{orig}}$ is the original MSA count. This design prevents evolutionary information distortion caused by excessive generated MSAs.

## 3 EXPERIMENT

**Baselines**  To evaluate PLAME's capability in generating high-fidelity and diverse MSAs, we compared it with several state-of-the-art AI-based MSA generation methods and AlphaFold2's MSA pipeline (Jumper et al., 2021). The baselines include AF2 MSA (Johnson et al., 2010), and open-source methods including EvoDiff and MSAGPT (Chen et al., 2024; Alamdari et al., 2023). Additionally, we include an MSA-free method, ESMFold (Lin et al., 2023), to evaluate the complementary benefits of explicit MSA enhancement versus implicit evolutionary modeling.

**Datasets**  For the training dataset, we used the PDB and UniClust30 subsets from the OpenProteinSet as our data source (Ahdritz et al., 2024b). The pre-searched MSAs from OpenFold training were also included. We retained data with at least 64 MSA sequences. To avoid overlap with the test cases, we removed sequences with over 90% similarity by MMSeqs based on UniClust30 clustering results (Mirdita et al., 2017; Steinegger & Söding, 2017). This process yielded an initial dataset of 293,979 samples, which were split into training and validation sets with a 90:10 ratio. For the test dataset, we adopted the curated test cases from MSAGPT (Chen et al., 2024), which consist of 200 protein samples from three benchmarks: CASP14&15, CAMEO (Haas et al., 2018), and PDB (Berman et al., 2000). Any $> 90\%$ redundancy between the test cases and training dataset was eliminated.

**Evaluation    Structural Assessment Metric** We evaluate structure qualitywith local and global metrics. Local metrics include pLDDT (per-residue confidence) and LDDT (local distance difference test). Global metrics comprise GDT (global distance test), TM-Score (template modeling score) (Zhang & Skolnick, 2005), pTM (predicted TM-score), and RMSD (root mean square deviation).

**AlphaFold2 Folding Modes** To comprehensively assess MSA augmentation effectiveness, we evaluate three AF2 configurations with increasing computational complexity:

- **Mode1**: pTM-3 model without templates (fast baseline) (Jumper et al., 2021)
- **Mode2**: Default 5 models without templates (standard setting) (Jumper et al., 2021)
- **Mode3**: Default 5 models with templates (full capability) (Jumper et al., 2021)
- **AF3**: Default 5 models with templates by AlphaFold3 (Abramson et al., 2024)

**Sequence Assessment Metric** We employ four sequence-based metrics to quantify alignment fidelity and diversity:

**1) Conservation Score** measures residue conservation at each position: $C_i = \text{Freq}_{\max}(i)/N$, where $\text{Freq}_{\max}(i)$ is the most frequent residue at position $i$ and $N$ is the sequence count. Higher scores indicate stronger evolutionary constraints.

**2) Gap Proportion** quantifies alignment completeness: $G_i = G(i)/N$, where $G(i)$ counts gaps at position $i$. Lower values indicate better alignment quality.

**3) Substitution Compatibility** evaluates evolutionary plausibility using BLOSUM62 scores $S_{\text{BLOSUM}}$ (Eq. 14). Higher scores reflect greater biological relevance.

**4) Alignment Entropy** captures positional diversity via Shannon entropy:

$$H_i = - \sum_{r \in \{R_i\}} p(r) \log_2 p(r) \tag{16}$$

where $\{R_i\}$ represents unique residues at position $i$ and $p(r) = \text{count}(r)/N$. Higher entropy indicates greater diversity; lower entropy suggests functional conservation.

## 3.1    Structure Benchmark Assessment

We evaluated PLAME across three AF2 folding modes and AF3, using six structural metrics to assess MSA generation quality (See details in Table1).

### 3.1.1    General Performance Comparison

PLAME demonstrates consistent superiority across both zero-shot and few-shot scenarios against traditional MSA searching, AI-based searching, and AI-based generative methods, establishing a new paradigm for MSA generation without traditional homology search. In zero-shot settings, where proteins lack existing MSAs, PLAME achieves remarkable improvements with pLDDT scores reaching 71.50 in Mode3, significantly outperforming competing methods like EvoDiff (64.39) and MSAGPT (68.39). Moreover, the performance gap becomes even more pronounced in challenging scenarios: while EvoDiff and MSAGPT often introduce detrimental noise when their generated sequences are concatenated with original AF2 MSAs, PLAME consistently enhances folding quality. Interestingly, few-shot scenarios reveal that existing methods can partially recover performance when guided by initial homologous sequences, yet PLAME maintains its edge by generating more coherent evolutionary profiles that complement rather than interfere with existing MSAs.

### 3.1.2    Mode-Dependent Performance Patterns

The progression from Mode1 through AF3 reveals intriguing insights about the relationship between model sophistication and MSA augmentation benefits. Mode1 and Mode2 demonstrate the strongest relative improvements from PLAME-generated MSAs, with pLDDT gains of up to 5 points across different baseline methods. As configurations advance to Mode3 with structural templates, the enhancement effects become more nuanced—while absolute performance continues to improve, the marginal gains from MSA augmentation diminish because template information already captures

Table 1: Performance metrics across different modes and models. The best results in each folding mode are highlighted in bold. Zero and Few indicate zero-shot (proteins without MSAs) and few-shot cases (proteins with existing MSAs), respectively.

| | pLDDT (↑) | | GDT (↑) | | TMscore (↑) | | RMSD(↓) | | LDDT (↑) | | pTM (↑) | |
|---|---|---|---|---|---|---|---|---|---|---|---|---|
| | Zero | Few | Zero | Few | Zero | Few | Zero | Few | Zero | Few | Zero | Few |
| ESMFold | 66.26 | 62.62 | 0.6 | 0.53 | 0.6 | 0.57 | 9.58 | 12.04 | 0.62 | 0.59 | / | / |
| **Mode1** | | | | | | | | | | | | |
| AF2 MSA | 60.07 | 62.14 | 0.50 | 0.52 | 0.50 | 0.57 | 12.34 | 12.16 | 0.54 | 0.58 | 0.44 | 0.49 |
| EvoDiff | 58.68 | 61.83 | 0.46 | 0.50 | 0.46 | 0.54 | 13.81 | 12.95 | 0.50 | 0.56 | 0.40 | 0.48 |
| MSAGPT | 59.81 | 61.18 | 0.48 | 0.51 | 0.48 | 0.56 | 12.62 | 12.35 | 0.53 | 0.57 | 0.43 | 0.48 |
| DHR | 63.64 | 62.60 | 0.51 | 0.52 | 0.52 | 0.57 | 12.04 | 11.92 | 0.55 | 0.59 | / | / |
| **PLAME** | **66.54** | **66.08** | **0.53** | **0.54** | **0.53** | **0.58** | **11.48** | **12.14** | **0.57** | **0.60** | **0.49** | **0.52** |
| **Mode2** | | | | | | | | | | | | |
| AF2 MSA | 66.56 | 66.32 | 0.51 | 0.55 | 0.52 | 0.60 | **12.06** | 11.84 | 0.55 | 0.61 | / | / |
| EvoDiff | 61.98 | 65.83 | 0.48 | 0.53 | 0.48 | 0.58 | 14.23 | **11.82** | 0.52 | 0.59 | / | / |
| MSAGPT | 64.88 | 65.96 | 0.51 | **0.56** | 0.51 | 0.60 | 12.60 | 11.90 | 0.55 | 0.61 | / | / |
| **PLAME** | **67.77** | **67.48** | **0.53** | 0.55 | **0.54** | **0.60** | 12.62 | 11.90 | **0.57** | **0.61** | / | / |
| **Mode3** | | | | | | | | | | | | |
| AF2 MSA | 70.31 | 69.61 | 0.57 | 0.60 | 0.57 | 0.64 | **10.53** | **10.24** | 0.60 | **0.65** | / | / |
| EvoDiff | 64.39 | 68.54 | 0.51 | 0.57 | 0.51 | 0.61 | 13.20 | 10.81 | 0.54 | 0.62 | / | / |
| MSAGPT | 68.39 | 69.30 | 0.57 | **0.60** | 0.56 | 0.64 | 11.05 | 10.40 | 0.59 | 0.64 | / | / |
| **PLAME** | **71.50** | **70.48** | **0.58** | 0.59 | **0.58** | **0.64** | 11.41 | 10.62 | **0.60** | 0.64 | / | / |
| **AF3** | | | | | | | | | | | | |
| AF2 MSA | 66.34 | **72.54** | 0.55 | 0.61 | **0.56** | 0.65 | 11.29 | 10.29 | 0.58 | **0.66** | / | / |
| **PLAME** | **70.23** | 72.00 | **0.55** | **0.62** | 0.55 | **0.65** | **11.23** | **10.26** | **0.59** | 0.65 | / | / |

substantial evolutionary constraints. This phenomenon reflects a fundamental trade-off in modern protein folding: as models become more powerful and incorporate diverse information sources, the additional value of synthetic MSAs decreases, though PLAME's high-quality generations continue to provide meaningful contributions. The AF3 results further validate this trend, showing that PLAME maintains its effectiveness even with more advanced folding architectures, suggesting that high-quality virtual MSAs remain valuable complements to cutting-edge structural prediction methods.

### 3.1.3 PLAME vs ESMFold: Bridging Efficiency and Accuracy

The comparison with ESMFold reveals PLAME's unique position in the protein folding landscape, offering a compelling alternative that combines computational efficiency with enhanced accuracy. While ESMFold achieves reasonable baseline performance (pLDDT of 66.26), PLAME progressively widens this gap as more sophisticated folding configurations are employed. In basic Mode1, PLAME shows modest improvements, but the advantage becomes substantial in Mode3 where PLAME reaches 71.50 pLDDT compared to ESMFold's unchanged 66.26. This trend suggests that PLAME-generated MSAs provide increasingly valuable evolutionary context that more advanced folding models can effectively exploit. The consistent RMSD improvements across all modes further validate that PLAME's virtual MSAs contribute meaningful structural constraints, enabling users to achieve AF2-level accuracy while maintaining the computational advantages of MSA-free approaches.

### 3.2 Sequence quality assessment

To evaluate generated MSA quality beyond structural perspectives, we conducted sequence-level analysis by fidelity and diversity metrics. This provides an additional critical gap—establishing criteria for understanding generated MSA quality. Figure 3 presents our comparative analysis.

**PLAME achieves superior evolutionary fidelity by closely mimicking the distributional characteristics of natural MSAs across all key metrics.** The results reveal PLAME's distributions align most closely with AF2 MSAs in Conservation Score, Gap Proportion, and Substitution Compatibility, demonstrating its ability to capture authentic evolutionary constraints.

This fidelity advantage manifests in higher Conservation Scores and Substitution Compatibility values, indicating that PLAME-generated sequences preserve functionally critical residues while incorporating biologically plausible substitutions. The significantly lower Gap Proportion validates PLAME's approach, as the evolutionary latent space from ESM-2 provides richer homology information enabling more complete alignments.

**PLAME maintains diversity levels comparable to natural AF2 MSAs, supporting our hypothesis that excessive diversity introduces detrimental noise.** Rather than maximizing diversity like EvoDiff, this measured approach aligns with our selection strategy principles, where balanced information enrichment proves more effective than naive sequence proliferation (Section 2.4). The findings suggest successful MSA generation requires maintaining the delicate balance between providing sufficient homologous information and avoiding noise from unconstrained sequence generation, positioning PLAME as a method that respects fundamental biological constraints.

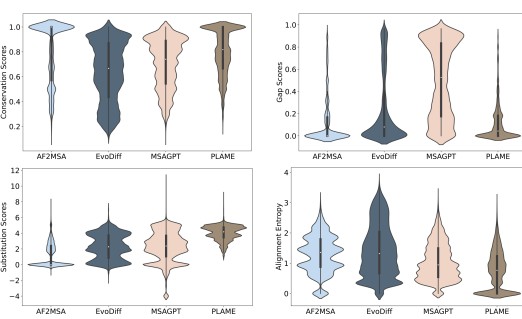

Figure 3: Comparison of sequence-based metrics for AF2 MSAs and MSAs generated by EvoDiff, MSAGPT, and PLAME.

Table 2: Ablation study of HiFiAD on PLAME and other baselines.

| | pLDDT (↑) | | GDT (↑) | | TMscore (↑) | | RMSD (↓) | | LDDT (↑) | | pTM (↑) | |
|---|---|---|---|---|---|---|---|---|---|---|---|---|
| | Zero | Few | Zero | Few | Zero | Few | Zero | Few | Zero | Few | Zero | Few |
| Random-16 | 63.61 | 62.63 | 0.52 | 0.51 | 0.52 | 0.56 | 12.01 | 12.67 | 0.55 | 0.58 | 0.46 | 0.49 |
| Blosum-8 | 61.04 | 62.71 | 0.5 | 0.52 | 0.51 | 0.57 | 12.53 | 12.69 | 0.55 | 0.58 | 0.45 | 0.50 |
| Blosum-32 | 62.97 | 62.40 | 0.50 | 0.50 | 0.51 | 0.55 | 12.28 | 12.84 | 0.55 | 0.57 | 0.45 | 0.48 |
| Top-Rec-16 | 62.04 | 62.93 | 0.51 | 0.51 | 0.51 | 0.55 | 12.15 | 12.48 | 0.55 | 0.57 | 0.45 | 0.49 |
| Top-down-Rec-16 | 63.43 | 63.10 | 0.52 | 0.52 | 0.51 | 0.57 | 11.97 | 12.15 | 0.55 | 0.58 | 0.46 | 0.49 |
| EvoDiff-HiFiAD | 58.24 | 60.89 | 0.46 | 0.49 | 0.46 | 0.54 | 13.74 | 12.39 | 0.51 | 0.56 | / | / |
| MSAGPT-HiFiAD | 60.16 | 62.63 | 0.48 | 0.52 | 0.48 | 0.57 | 12.54 | 12.18 | 0.53 | 0.59 | / | / |
| DHR-HiFiAD | 66.01 | 66.08 | 0.53 | 0.55 | 0.53 | 0.60 | 11.48 | 12.14 | 0.57 | 0.60 | / | / |
| **PLAME-HiFiAD** | **66.54** | **66.08** | **0.53** | **0.54** | **0.53** | **0.58** | **11.48** | **12.14** | **0.57** | **0.60** | **0.49** | **0.52** |

## 3.3 ABLATION STUDIES

To validate our HiFiAD selection strategy, we conducted ablation experiments across different selection approaches and baseline methods. Table 2 compares various selection strategies and evaluates HiFiAD's effectiveness on other generative methods.

**HiFiAD consistently outperforms alternative selection strategies by optimally balancing fidelity and diversity constraints.** Compared to similarity-based methods (Top/Down-Rec) and substitution matrix approaches (BLOSUM-32), HiFiAD achieves superior performance with pLDDT scores of 66.54 in zero-shot settings, demonstrating the importance of jointly considering evolutionary fidelity and controlled diversity. The strategy effectively identifies high-fidelity sequences while maintaining sufficient diversity to prevent overly deterministic conservation patterns. HiFiAD automatically adapts to varying MSA quality levels and shot configurations, making it robust without requiring manual parameter tuning.

When applied to competing baselines, HiFiAD consistently improves performance: EvoDiff benefits from a 58.24 to 60.89 pLDDT improvement, MSAGPT gains from 60.16 to 62.63, and DHR advances from 66.01 to 66.08. These improvements demonstrate that HiFiAD addresses fundamental challenges in MSA selection across all generative approaches. The consistent gains across different generation paradigms—from diffusion-based (EvoDiff) to autoregressive (MSAGPT) and retrieval-based (DHR) methods—validate that the fidelity-diversity trade-off represents a universal principle in MSA augmentation. The improvement margins correlate with baseline method quality, suggesting HiFiAD provides proportional benefits while maintaining relative performance hierarchy.

Also, we conducted ablation study on MSA length (See Table3). PLAME shows overall improvement on all length ranges, where performs the largest improvement on 100-300 range. We believe this is because the MSA training data are mainly concentrated in this range (Chen et al., 2024).

Table 3: Ablation on protein length.

| | Length Range | pLDDT($\uparrow$) | GDT($\uparrow$) | TMscore($\uparrow$) | RMSD($\downarrow$) | LDDT($\uparrow$) |
|---|---|---|---|---|---|---|
| **AF2 MSA** | <100 | 71.03 | 0.64 | 0.52 | 7.77 | 0.61 |
| **AF2 MSA** | 100-300 | 59.50 | 0.49 | 0.53 | 12.46 | 0.54 |
| **AF2 MSA** | >300 | 56.29 | 0.43 | 0.51 | 15.67 | 0.53 |
| **PLAME** | <100 | 74.12 | 0.63 | 0.52 | 7.49 | 0.61 |
| **PLAME** | 100-300 | 65.55 | 0.53 | 0.58 | 11.58 | 0.58 |
| **PLAME** | >300 | 58.31 | 0.45 | 0.53 | 16.16 | 0.54 |

Furthermore, we provide additional case studies on folding enhancement. More case studies on orphan *de novo* proteins (SectionE.5), protein failure cases (SectionE.4), selected protein cases with aligned structures (SectionE) can be found in the appendix.

pdb_id: 8okh_B

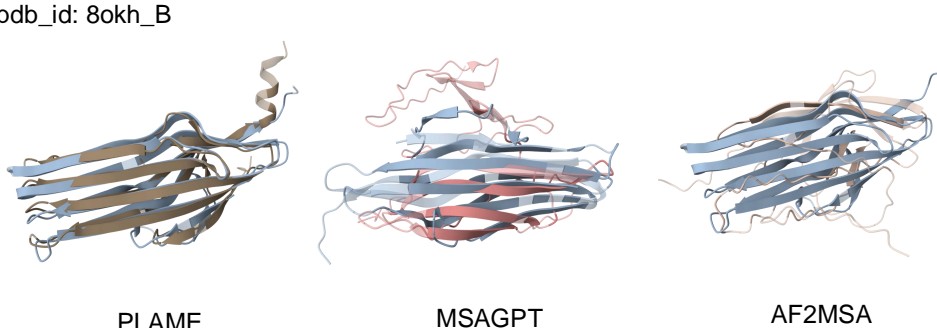

PLAME

pLDDT: 69.67
TMscore:  0.812
RMSD: 2.774

MSAGPT

pLDDT: 32.02
TMscore:  0.205
RMSD: 19.49

AF2MSA

pLDDT: 28.67
TMscore:  0.198
RMSD: 21.09

Figure 4: Case study of folding enhancement of PLAME, MSAGPT, and AF2 MSA on 8okh_B.

## 4 CONCLUSION

In this study, we introduce PLAME, the first model to leverage evolutionary embeddings for MSA generation and apply it to protein folding enhancement. Our approach bridges the gap between single-sequence inference and MSA-based methods, effectively improving protein folding performance. Evaluation results demonstrate that PLAME-generated MSAs outperform existing methods in both conservation and diversity metrics, significantly enhancing structural prediction accuracy across different protein families. PLAME serves as both an MSA enhancer and an efficient AlphaFold adapter without requiring time-consuming MSA searches, providing a fast, accurate, and scalable protein structure prediction solution. Additionally, our proposed quality metrics and experiments offer new insights into the relationship between MSA features and folding performance.

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

# A  PROOF OF THEOREM

We provide additional statements to demonstrate the superiority of the Conservation-Diversity Training Loss. Firstly, we demonstrate that the PCE Loss as a conservation-aware weighted loss by position in the perspective of MSA profiles.

**Lemma 1.** *Let $P(l, a)$ be the empirical amino–acid distribution for residue $a \in \mathcal{A}$, and let $Q_\theta(l, a)$ denote the model distribution at the residue (i.e. the conditional probability $p_\theta(a \mid y_{<l})$ after taking expectation over prefixes). Assign each column a weight $w_l \in [1 - \delta, 1 + \delta]$ obtained from its conservation score. Then PCE loss directs optimization preferentially toward conserved positions by minimizing a weighted KL divergence and scaling gradient magnitudes in proportion to $w_l$.*

*Proof.* For a sufficiently large set of $N$ homologous sequences sampled from $P$, the expected cross-entropy loss is

$$\mathbb{E}[\mathcal{L}_{\text{CE}}] = -\sum_{l=1}^{L} \sum_{a \in \mathcal{A}} P(l, a) \log Q_\theta(l, a). \tag{17}$$

Re-expressing each column term as $-\sum_a P \log Q = H\big(P(l, \cdot)\big) + D_{\text{KL}}\big(P(l, \cdot) \| Q_\theta(l, \cdot)\big)$, we obtain

$$\mathbb{E}[\mathcal{L}_{\text{CE}}] = \sum_{l=1}^{L} D_{\text{KL}}\big(P(l, \cdot) \| Q_\theta(l, \cdot)\big) + \sum_{l=1}^{L} H\big(P(l, \cdot)\big). \tag{18}$$

For the PCE loss,

$$\mathbb{E}[\mathcal{L}_{\text{PCE}}] = -\sum_{l=1}^{L} w_l \sum_{a \in \mathcal{A}} P(l, a) \log Q_\theta(l, a), \tag{19}$$

which can analogously be rewritten as the position-wise weighted KL

$$\mathbb{E}[\mathcal{L}_{\text{PCE}}] = \sum_{l=1}^{L} w_l D_{\text{KL}}\big(P(l, \cdot) \| Q_\theta(l, \cdot)\big) + \sum_{l=1}^{L} w_l H\big(P(l, \cdot)\big). \tag{20}$$

Let $\theta$ denote the model parameters. The gradient of the CE loss for column $l$ is

$$\frac{\partial \mathcal{L}_{\text{PCE}, l}}{\partial \theta} = -\sum_{a \in \mathcal{A}} P(l, a) \frac{1}{Q_\theta(l, a)} \frac{\partial Q_\theta(l, a)}{\partial \theta}. \tag{21}$$

For PCE the gradient is simply scaled by $w_l$:

$$\frac{\partial \mathcal{L}_{\text{PCE}, l}}{\partial \theta} = -w_l \sum_{a \in \mathcal{A}} P(l, a) \frac{1}{Q_\theta(l, a)} \frac{\partial Q_\theta(l, a)}{\partial \theta} = w_l \frac{\partial \mathcal{L}_{\text{CE}, l}}{\partial \theta}. \tag{22}$$

Consequently, in highly conserved columns the gradient magnitude is amplified by $1 + \delta$, whereas in variable columns ($w_l \approx 1 - \delta$) it is attenuated, focusing optimization effort on conserved regions.  $\square$

Based on the understanding of the PCE Loss, we then demonstrate that PCE Loss is expected to capture evolutionary information (MSA profile) with less error–measured by KL-Divergence.

**Theorem 1.** *Let $P(l, a)$ be the true amino–acid distribution in column $l$ ($l = 1, \ldots, L$) of an MSA and let $Q_\theta(l, a)$ be the distribution produced by a parametrised generative model $Q_\theta$. Denote the column–wise Kullback–Leibler divergence by*

$$D_{\text{KL}}\big(P(l, \cdot) \| Q_\theta(l, \cdot)\big) = \sum_{a \in \mathcal{A}} P(l, a) \log \frac{P(l, a)}{Q_\theta(l, a)}. \tag{23}$$

*Let*

$$\theta_{\text{CE}}^\star = \arg\min_\theta \mathcal{L}_{CE}(\theta), \qquad \theta_{\text{PCE}}^\star = \arg\min_\theta \mathcal{L}_{PCE}(\theta). \tag{24}$$

*Define the* average profile KL divergence

$$D_{KL}^{avg}(\theta) := \frac{1}{L} \sum_{l=1}^{L} D_{\mathrm{KL}}\big(P(l,\cdot) \,\|\, Q_\theta(l,\cdot)\big). \tag{25}$$

*Under the assumption that both optimization problems are solved to global optimality, the model trained with PCE Loss captures the MSA profile with less divergence $D_{KL}^{avg}$:*

$$D_{KL}^{avg}(\theta_{\mathrm{PCE}}^\star) \;\leq\; D_{KL}^{avg}(\theta_{\mathrm{CE}}^\star) \tag{26}$$

*Proof.* Rewrite two losses in the form of KL-Divergence $\sum_a P \log Q \;=\; H\big(P(l,\cdot)\big) + D_{\mathrm{KL}}\big(P(l,\cdot)\|Q_\theta(l,\cdot)\big)$, we have:

$$\mathcal{L}_{\mathrm{CE}}(\theta) = C_0 + \sum_{l=1}^{L} D_{\mathrm{KL}}\big(P(l,\cdot)\|Q_\theta(l,\cdot)\big),$$

$$\mathcal{L}_{\mathrm{PCE}}(\theta) = C_w + \sum_{l=1}^{L} w_l \, D_{\mathrm{KL}}\big(P(l,\cdot)\|Q_\theta(l,\cdot)\big), \tag{27}$$

where $C_0 = \sum_l H(P(l,\cdot))$ and $C_w = \sum_l w_l H(P(l,\cdot))$ are constants independent of $\theta$. Hence minimizing $\mathcal{L}_{\mathrm{PCE}}$ is equivalent to minimizing the *weighted* KL

$$D_w(\theta) := \sum_{l=1}^{L} w_l \, D_{\mathrm{KL}}\big(P(l,\cdot)\|Q_\theta(l,\cdot)\big), \qquad \theta_{\mathrm{PCE}}^\star = \arg\min_\theta D_w(\theta). \tag{28}$$

Then, since every $w_l$ is bounded, we can establish the relations:

$$(1-\delta) \sum_{l=1}^{L} D_{\mathrm{KL}}\big(P(l,\cdot)\|Q_\theta(l,\cdot)\big) \;\leq\; D_w(\theta) \;\leq\; (1+\delta) \sum_{l=1}^{L} D_{\mathrm{KL}}\big(P(l,\cdot)\|Q_\theta(l,\cdot)\big). \tag{29}$$

Dividing by $L$ gives:

$$(1-\delta) \, D_{\mathrm{KL}}^{\mathrm{avg}}(\theta) \;\leq\; \frac{D_w(\theta)}{L} \;\leq\; (1+\delta) \, D_{\mathrm{KL}}^{\mathrm{avg}}(\theta). \tag{$*$}$$

Based on the fact that $\theta_{\mathrm{PCE}}^\star$ minimizes $D_w$, denote $\Delta_w := D_w(\theta_{\mathrm{CE}}^\star) - D_w(\theta_{\mathrm{PCE}}^\star) \geq 0$. By applying $(*)$ to both optimal parameters and subtracting, we obtain:

$$(1-\delta)\Big[D_{\mathrm{KL}}^{\mathrm{avg}}(\theta_{\mathrm{CE}}^\star) - D_{\mathrm{KL}}^{\mathrm{avg}}(\theta_{\mathrm{PCE}}^\star)\Big] \;\leq\; \frac{\Delta_w}{L}. \tag{30}$$

Since $\Delta_w \geq 0$ and $1-\delta > 0$; it is strictly positive whenever $\Delta_w > 0$, Therefore,

$$D_{\mathrm{KL}}^{\mathrm{avg}}(\theta_{\mathrm{PCE}}^\star) \;\leq\; D_{\mathrm{KL}}^{\mathrm{avg}}(\theta_{\mathrm{CE}}^\star), \tag{31}$$

which completes the proof. $\qquad\square$

A natural challenge emerges when applying the PCE Loss—the model tends to accurately capture the distribution of conserved regions while neglecting the distribution of variable regions. To address this issue, we demonstrate that the DIRE Loss effectively enhance the modeling in the variable regions.

**Theorem 2.** *For $l = 1, \ldots, L$ let $P(l,a)$ denote the empirical amino-acid distribution and $Q_\theta(l,a)$ any model. When each amnio acid site is optimized independently, the minimizer is*

$$Q_\alpha^\star(l,a) = \frac{P(l,a)^{\tau_l}}{\sum_{b\in\mathcal{A}} P(l,b)^{\tau_l}}, \qquad \tau_l = \frac{\alpha w_l}{\alpha w_l + (1-\alpha)} \in (0,1). \tag{32}$$

*Moreover,*

$$H\big(P(l,\cdot)\big) \leq H\big(Q_\alpha^\star(l,\cdot)\big) \leq \log|\mathcal{A}|, \tag{33}$$

*with the entropy increase largest when $w_l$ is small (variable regions). Thus $\mathcal{L}_{DIRE}$ counter-acts the entropy suppression of $\mathcal{L}_{PCE}$ and serves as a principled regularizer on variable regions.*

*Proof.* Since the combined loss $\mathcal{L}_\alpha$ sums over amino acid positions, we may analyze a single site independently, denoting $P(a) = P(l, a)$, $Q(a) = Q(l, a)$ and $w = w_l$. For each site we minimize, we have

$$F(Q) = \alpha w \sum_a P(a) \log \frac{P(a)}{Q(a)} + (1 - \alpha) \sum_a Q(a) \log Q(a), \tag{34}$$

subject to the normalization constraint $\sum_a Q(a) = 1$.

Introducing a Lagrange multiplier $\lambda$ and setting the derivative with respect to $Q(a)$ to zero yields

$$-\frac{\alpha w P(a)}{Q(a)} \; + \; (1 - \alpha)\big(1 + \log Q(a)\big) \; + \; \lambda \; = \; 0. \tag{35}$$

Solving this equation reveals a "temperature-like" solution based on $\tau$:

$$Q(a) \; \propto \; P(a)^\tau, \qquad \tau \; = \; \frac{\alpha w}{\alpha w + (1 - \alpha)} \in (0, 1), \tag{36}$$

which is exactly the optima $Q_\alpha^\star(l, \cdot)$ mentioned earlier.

Since $0 < \tau < 1$, this transformation always increases entropy unless $P$ is already uniform:

$$H\big(P(l, \cdot)\big) \; \leq \; H\big(Q_\alpha^\star(l, \cdot)\big) \; \leq \; \log |\mathcal{A}|. \tag{37}$$

The entropy gain is larger when $w$ is small (in the variable regions). Consequently, the $(1 - \alpha)$, $\mathcal{L}_{\text{DIRE}}$ term counteracts the over-confidence induced by $\mathcal{L}_{\text{PCE}}$ in variable regions, serving as an adaptive entropy-based regularizer. $\qquad\square$

## B  TRAINING AND SAMPLING DETAILS

**Training Details**   We trained our model based on a Transformer T5 architecture, incorporating axial attention and task-specific modifications to enhance performance. The model consists of 12 encoder layers and 12 decoder layers, with a hidden size of 1024, 12 attention heads, and a feedforward dimension of 2048. The feedforward projection employs a gated-GELU activation function. During training, we employed the AdamW optimizer with a learning rate of 5e-5, a weight decay of 1e-5, and a polynomial decay scheduler with a 1% warmup ratio. Training was conducted on four NVIDIA A40 GPUs for up to 200,000 steps, with a batch size of 4 per device for both training and evaluation.

**Sampling details**   The sampling process was configured with the following parameters: we generate 16 MSAs for 4 trials per generation. The sampling used a repetition penalty of 1.0, a temperature of 1.0, and top-p sampling with a threshold of 0.95. Beam search was performed with 4 beams and 1 beam group. Sampling was executed on an A40 GPU.

## C  RELATED WORKS

**Protein Structure Prediction**   Protein structure prediction methods fall into three main categories: physics-based, homology-based, and deep learning approaches. Physics-based methods, such as AMBER and CHARMM, use molecular physics and energy optimization to simulate protein folding (Cornell et al., 1995; Brooks et al., 2009). While offering detailed folding insights, they are computationally expensive and sensitive to initial conditions, often yielding suboptimal results (Karplus & McCammon, 2002; Freddolino et al., 2010; Pande et al., 2010). Homology modeling tools, like Rosetta and HHpred, use MSAs and evolutionary data to predict structures by refining templates from known experimental structures (Rohl et al., 2004; Hildebrand et al., 2009). These methods perform well with suitable templates but struggle with orphan proteins and low-homology families (Webb & Sali, 2016; Baker & Sali, 2001). Deep learning-based methods, such as AlphaFold2 and OmegaFold, use advanced neural architectures and protein templates to achieve near-experimental accuracy with greater speed and scalability (Jumper et al., 2021; Abramson et al., 2024; Wu et al., 2022). Despite their success, they still depend on high-quality MSAs and struggle with low-homology proteins.

**AlphaFold-based Enhancement**   Building on AlphaFold's success, researchers have developed methods to refine specific modules, aiming to improve accuracy or efficiency. These advancements can be grouped into three main categories. The first category focuses on homology expansion techniques, such as MMSeq2 and DeepMSA2, which expand the evolutionary search space to enhance prediction accuracy. However, these methods often slow down inference despite their modest performance gains (Johnson et al., 2010; Steinegger & Söding, 2017; Zheng et al., 2024; Lee et al., 2024). The second category targets search acceleration, with methods like ColabFold and ESMFold bypassing the MSA search process to enhance computational efficiency. However, this speedup often results in incomplete evolutionary data, potentially reducing prediction accuracy (Lin et al., 2023; Mirdita et al., 2022). The third category leverages generative models to capture protein homology and augment input data, especially for orphan proteins and low-homology families. While promising in specific scenarios, these models struggle with extremely limited evolutionary signals, and their artificial sequences often deviate from traditional MSA distributions, limiting broader applicability (Alamdari et al., 2023; Zhang et al., 2022; 2023; Chen et al., 2024).

## D   COMPARISON ON INFERENCE SPEED AND MEMORY USAGE

To further demonstrate PLAME's efficiency, we calculated the inference time and memory cost of each method. We used ENZYME 1.2.1.50 (EC Number) with length 488 as the test case. The results show that PLAME achieved the fastest speed among all AI-based methods while consuming only 4.5GB of memory. The processing speed is comparable to traditional methods like MMSeq2 and AI-based retrieval methods like DHR. Compared to retrieval-based methods, PLAME does not require downloading or building databases in advance, nor does it need preprocessing steps. This makes it more lightweight and efficient for deployment.

| Method | Time per MSA (s) | GPU Memory (Gb) |
|---|---|---|
| **PLAME** | 0.10 | 4.5 |
| **DHR** | 0.16 + 358.61 (Alignment) | 1.9 |
| **MMSeq2** | 0.48 | 0.0 |
| **MSAGPT** | 62.46 | 41.6 |
| **EvoDiff** | 478.24 | 4.0 |

Table 4: Comparison on inference speed and memory.

## E   EXTENSIVE CASE STUDIES

### E.1   CASE STUDY ON SUCCESSFUL DESIGNS

To further explore the key pattern of the MSA augmentation, we provide a series of sequence and structure visualization in Appendix H. We select representative cases collected from different datasets and range from different lengths to comprehensive evaluate the samples.

Among these cases, we can generally observe that most generated MSA sequences maintain high similarity with the query sequence. Furthermore, the generated MSAs provide good enhancement at the originally conserved sites. This indicates that protein language models can still retain some evolutionary information even for proteins with low homology, although the diversity they can provide is more limited due to homology constraints.

Additionally, we identified several patterns in the sampled MSAs that clearly deviate from the original distribution, such as consecutive gaps (in 8ehb_F), repeated HHHHHH sequences (in 8okw_B), and repeated SSSSSSSSS (in 7xrl_A). We believe these erroneous generations are related to the autoregressive generation method, where the model tends to produce excessive hallucinations after getting trapped in incorrect local probability distributions. We also observed that these failure patterns occur more frequently in longer sequences, possibly due to insufficient training on cases with greater length. These represent an area requiring further improvement.

## E.2 Folding Enhancement on Average Proteins

To probe the effectiveness of PLAME on average proteins, we firstly build a dataset from PDB validation set with 36 proteins. These protein MSAs don't have sequence similarity over 90% compared to the PLAME training set. We randomly employ 32 MSAs for each protein and augment them with designed MSAs after HiFiAD filtering. The results are shown in Table 5. From the

|          | pLDDT  | GDT   | TMscore | RMSD  | LDDT  | pTM   |
|----------|--------|-------|---------|-------|-------|-------|
| **AF2 MSA** | 83.156 | 0.767 | 0.785   | 5.243 | 0.753 | 0.718 |
| **PLAME**   | 83.328 | 0.775 | 0.795   | 5.028 | 0.757 | 0.723 |

Table 5: Comparison of folding enhancement on average proteins

experimental results, the effects of augmentation align with our initial assumptions, demonstrating modest improvements. While the overall topological structure remains unchanged, minor adjustments can be observed in the structural details. As reported in MSAGPT, performance gains approach saturation between 16 and 32 augmentations. The relatively small improvements observed when applying our method to the average protein MSA can be attributed to the fact that these original MSAs already provide sufficient evolutionary information to AlphaFold2's MSA Transformer, thus limiting the potential impact of additional augmentation.

## E.3 Further Ablation on MSA Filtering

We further validate the effectiveness of filtered high-quality MSAs by comparing the performance with the more randomly selected MSAs (64 for each protein). From Table 6 and 2, We can observe a

|                    | pLDDT  | GDT   | TMscore | RMSD   | LDDT  | pTM   |
|--------------------|--------|-------|---------|--------|-------|-------|
| **More Random MSAs** | 63.620 | 0.512 | 0.533   | 12.692 | 0.563 | 0.473 |
| **HiFiAD**           | 66.349 | 0.534 | 0.553   | 11.755 | 0.581 | 0.506 |

Table 6: Comparison of folding enhancement based on different filterings.

slight performance enhancement compared to Random-16 filtering approach according to pLDDT and LDDT. Conversely, the performance on global metric decreases. From the results, more co-evolutionary information may lead to better local geometric conformation, but it will disturb the modeling of the global conformations due to the bias during generation.

## E.4 Failure Case Analysis

Other than analyzing successful cases, we analyzed four representative failure cases (3bog_B, 7sxb_A, 8gzu_AN, 8gzu_T3) with the largest performance drops, which includes three zero-shot and one few-shot examples. From the detailed results, we observe a clear mismatch between global metric, including GDT, TMScore, and RMSD, and local metric, including pLDDT, LDDT, and pTM on 3bog_B and 8gzu_T3. It is consistent with the metric discrepancies we observed in the main experiment.

Among the visualized MSA cases, we observed that generated MSAs contained extremely similar sequences (>90% similarity). Specifically, these high-similarity sequences caused all sites to appear more conserved, resulting in a lack of covariation patterns necessary for AlphaFold2 to infer structural contacts. This pattern was evident across all four cases. Notably, for 3bog_B and 8gzu_T3, the generated high-similarity MSAs further enhanced the conservation of already conserved regions, which consequently led to improvements in global metrics.

## E.5 De novo Protein Folding Enhancement

We conduct further experiments on De Novo protein cases, where almost of them are orphan. Examples of de novo proteins include 8SK7 (RFDiffusion (Watson et al., 2023)), 8TNM/8TNO

|           | pLDDT  | GDT   | TMscore | RMSD   | LDDT  | pTM   |
|-----------|--------|-------|---------|--------|-------|-------|
| **AF2 MSA** |||||||
| **3bog_B**  | 41.493 | 0.150 | 0.130   | 22.443 | 0.148 | 0.129 |
| **7sxb_A**  | 84.931 | 0.739 | 0.757   | 2.559  | 0.661 | 0.753 |
| **8gzu_AN** | 58.189 | 0.390 | 0.488   | 17.630 | 0.700 | 0.406 |
| **8gzu_T3** | 59.533 | 0.591 | 0.668   | 14.030 | 0.659 | 0.597 |
| **PLAME** |||||||
| **3bog_B**  | 32.918 | 0.169 | 0.148   | 17.522 | 0.158 | 0.118 |
| **7sxb_A**  | 53.956 | 0.358 | 0.358   | 9.988  | 0.369 | 0.359 |
| **8gzu_AN** | 51.542 | 0.393 | 0.491   | 17.238 | 0.513 | 0.414 |
| **8gzu_T3** | 55.169 | 0.377 | 0.480   | 20.930 | 0.691 | 0.394 |

Table 7: Comparison of folding enhancement on failure cases.

(Chroma (Ingraham et al., 2023)), and 8CYK (ProteinMPNN (Dauparas et al., 2022)). We followed the same augmentation pattern as the main experiment. From Table 8, we observed that PLAME

|             | pLDDT | GDT   | TMscore | RMSD  | LDDT  | pTM   |
|-------------|-------|-------|---------|-------|-------|-------|
| **AF2 MSA** | 89.27 | 0.886 | 0.904   | 1.658 | 0.781 | 0.800 |
| **HiFiAD**  | 88.33 | 0.924 | 0.940   | 1.483 | 0.824 | 0.800 |

Table 8: Comparison of folding enhancement on de novo proteins.

experiences a slight decrease in pLDDT scores while simultaneously showing improvements in other metrics. The generated MSA visualizations in Figures 5 and 6 reveal that most generated sequences maintain $> 70\%$ similarity to the query sequences. This phenomenon may be attributed to these test cases being highly Out-Of-Distribution (OOD) relative to the training dataset. Nevertheless, the diverse sampling strategy still effectively enhances the profile information of orphan proteins, resulting in substantial performance improvements. Furthermore, we visualized specific local regions where PLAME achieves superior alignment performance as measured by TMscore. Analysis revealed that across all augmented profiles, these high-performing local regions exhibit remarkable conservation, suggesting a strong correlation between sequence conservation patterns and structural alignment quality.

# F DISCUSSION

## F.1 LIMITATIONS

Recent advancements in MSA generation models have shown promising results in enhancing protein folding predictions. However, several challenges remain to be addressed for broader applications and improved performance. **1) Limited quality** by current model architectures, data constraints, and generation strategies, such as relying on small MSA prompts, hinders the overall richness and informativeness of the generated MSAs. Future methods should focus on constructing more expressive evolutionary latent spaces to better capture the complexity of protein sequence relationships and improve the informativeness of generated MSAs. **2) Distribution gaps** still exist between the diversity and quality of generated MSAs and their natural counterparts, limiting their utility in broader applications. While current methods show potential in folding tasks, future models should focus on zero-shot generation capabilities to produce MSAs with distributions closer to natural MSAs, enabling broader applications such as conserved residue identification, mutation effect prediction, and functional annotation. **3) Assessing MSA quality** remains an unresolved issue, as current evaluations primarily rely on downstream folding performance to infer quality. Developing direct and robust quality assessment metrics will be crucial for systematically evaluating and improving MSA generation methods, enabling the selection of high-quality MSAs for specific applications and paving the way for next-generation models with enhanced accuracy, broader applicability, and greater biological relevance.

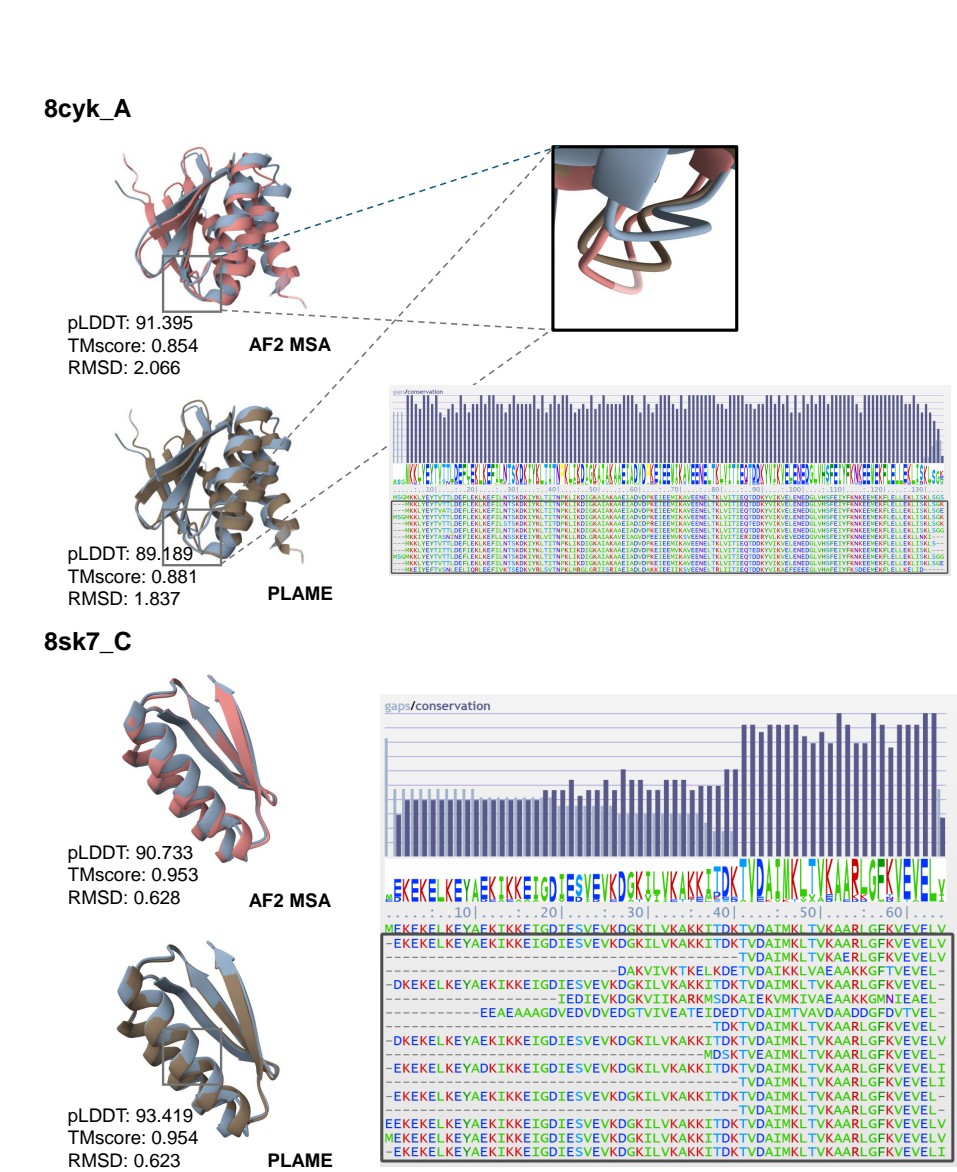

Figure 5: Comparison of structure enhancement of De Novo proteins.

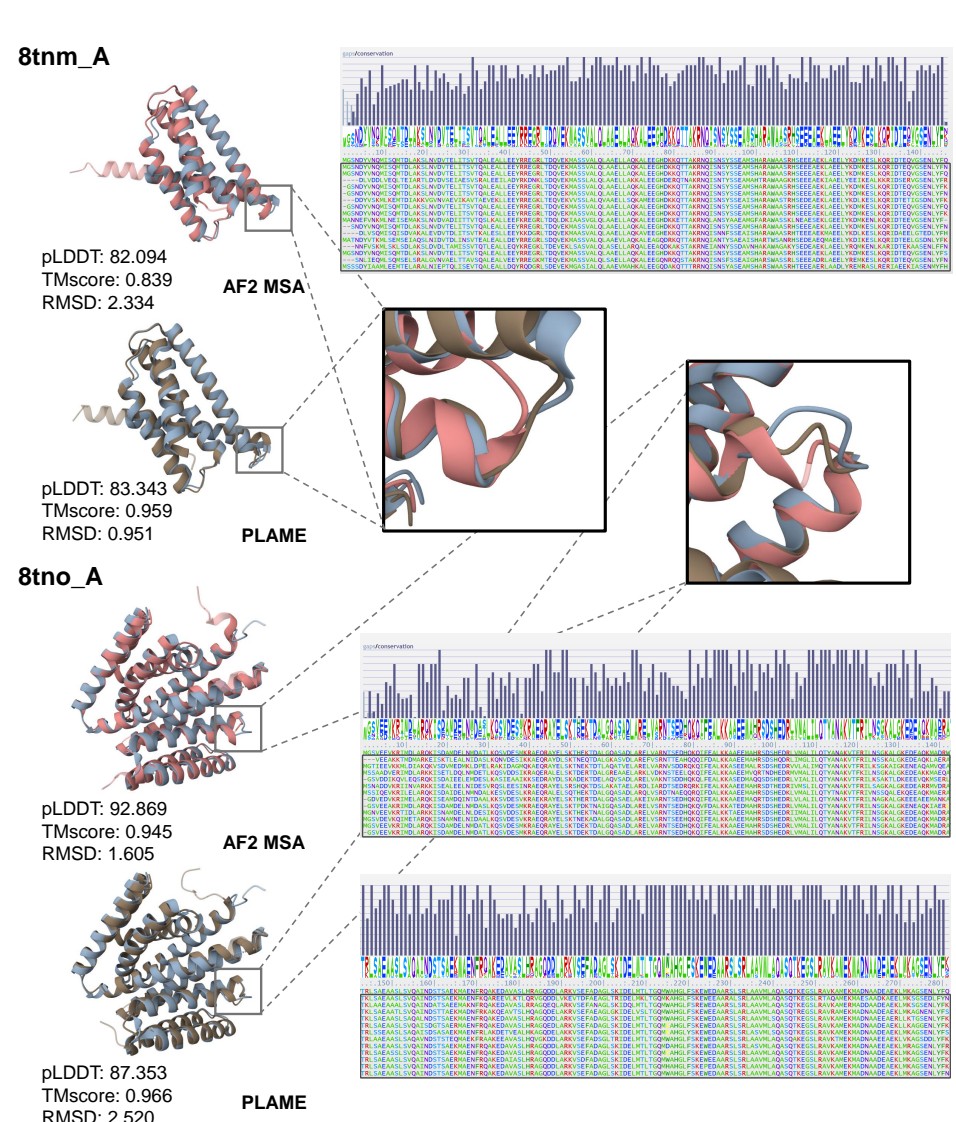

Figure 6: Comparison of structure enhancement of De Novo proteins.

## G    STRUCTURE COMPARISON VISUALIZATION

pdb_id: 8ehb_F

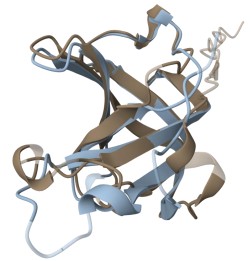 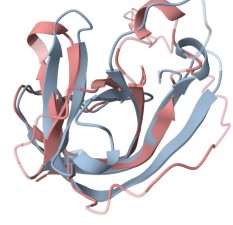 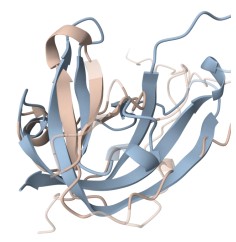

PLAME

pLDDT: 75.64
TMscore: 0.749
RMSD: 3.218

MSAGPT

pLDDT: 41.35
TMscore: 0.563
RMSD: 4.462

AF2MSA

pLDDT: 36.25
TMscore: 0.359
RMSD: 9.653

Figure 7: Structure comparison visualization of 8ehb_F.

pdb_id: 8b4k_C

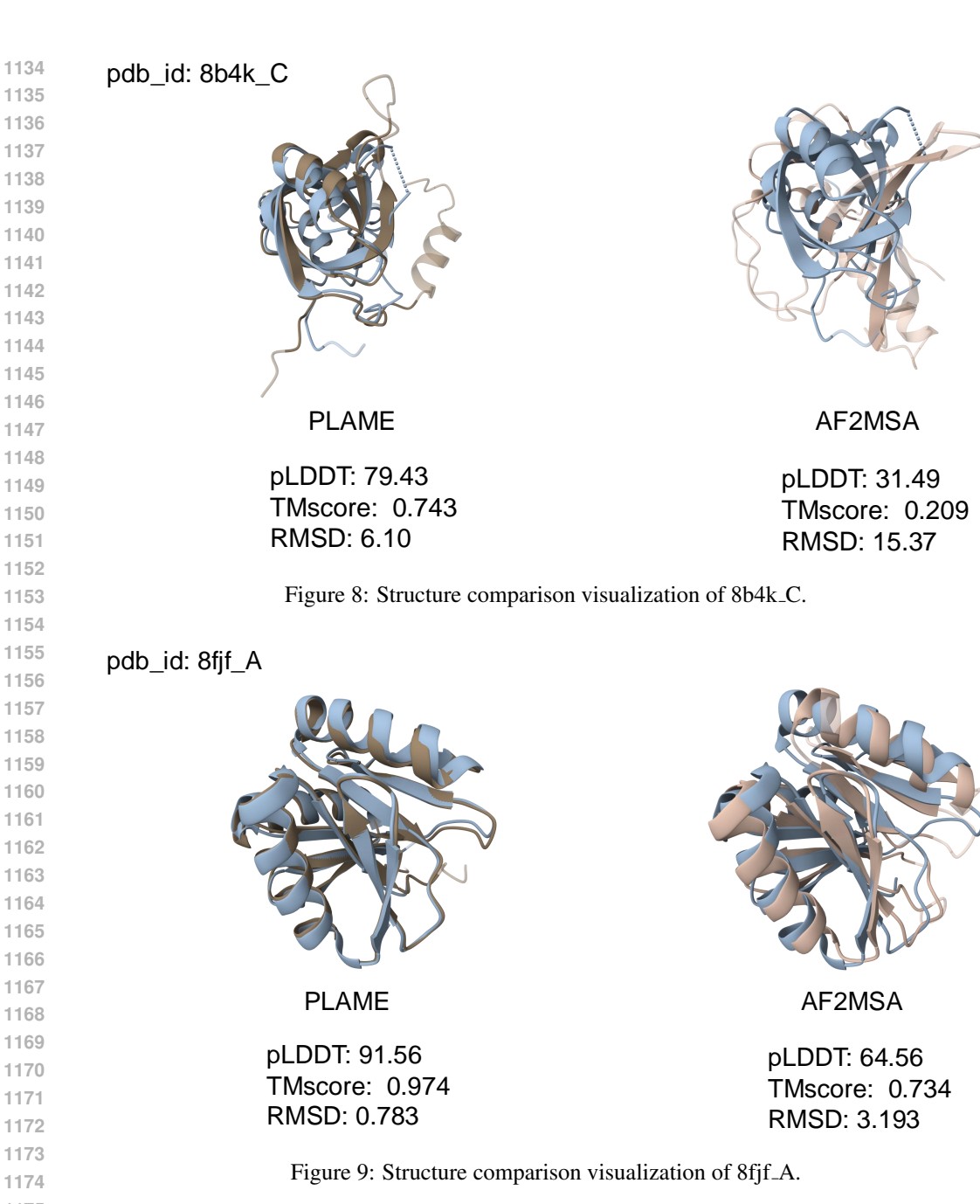

PLAME

pLDDT: 79.43
TMscore: 0.743
RMSD: 6.10

AF2MSA

pLDDT: 31.49
TMscore: 0.209
RMSD: 15.37

Figure 8: Structure comparison visualization of 8b4k_C.

pdb_id: 8fjf_A

PLAME

pLDDT: 91.56
TMscore: 0.974
RMSD: 0.783

AF2MSA

pLDDT: 64.56
TMscore: 0.734
RMSD: 3.193

Figure 9: Structure comparison visualization of 8fjf_A.

pdb_id: 8eoz_B

PLAME

pLDDT: 88.24
TMscore: 0.958
RMSD: 0.127

MSAGPT

pLDDT: 48.10
TMscore: 0.290
RMSD: 15.338

Figure 10: Structure comparison visualization of 8eoz_B.

pdb_id: 8okw_B

PLAME

pLDDT: 86.18
TMscore: 0.945
RMSD: 1.408

MSAGPT

pLDDT: 52.90
TMscore: 0.658
RMSD: 12.459

Figure 11: Structure comparison visualization of 8okw_B.

## H  AUGMENTED MSA VISUALIZATION

To provide an intuitive understanding of the MSAs generated by PLAME, we selected several representative cases for visualization. These cases demonstrate consistent improvements in folding accuracy compared to the MSAs provided by AF2 and cover a range of sequence lengths, including short (<100), medium (100-300), and long (>300) sequences, as well as cases under few-shot and zero-shot settings. For each visualization, the generated MSAs are highlighted with a black box. Additionally, the upper portion of each figure presents conservation information alongside the corresponding gap information. The protein information is provided in the left-top corner at each figure.

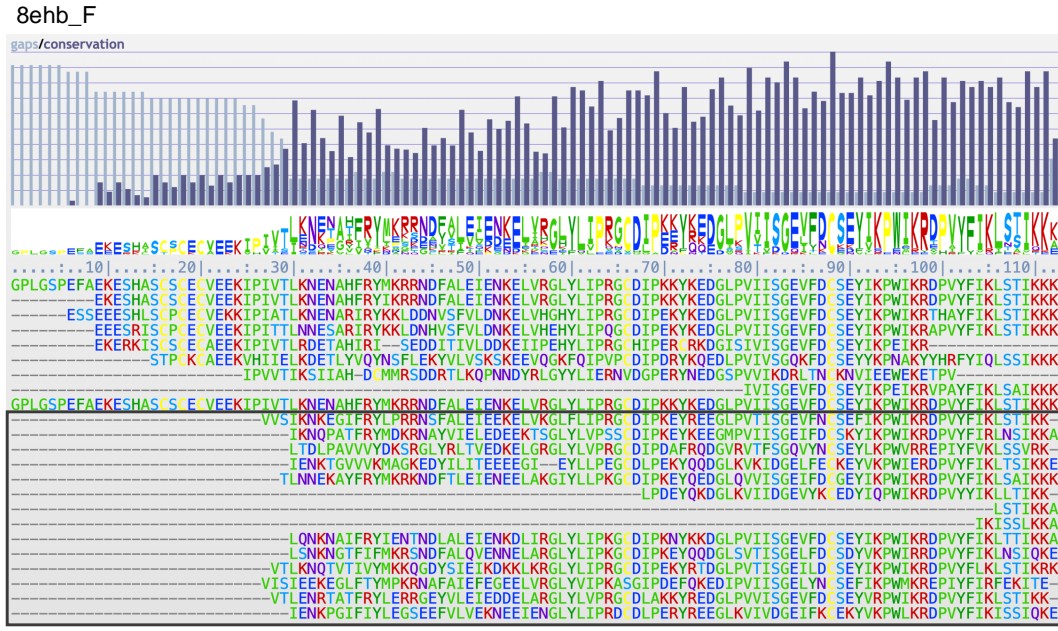

Figure 12: Augmented MSA visualization of 8ehb_F.

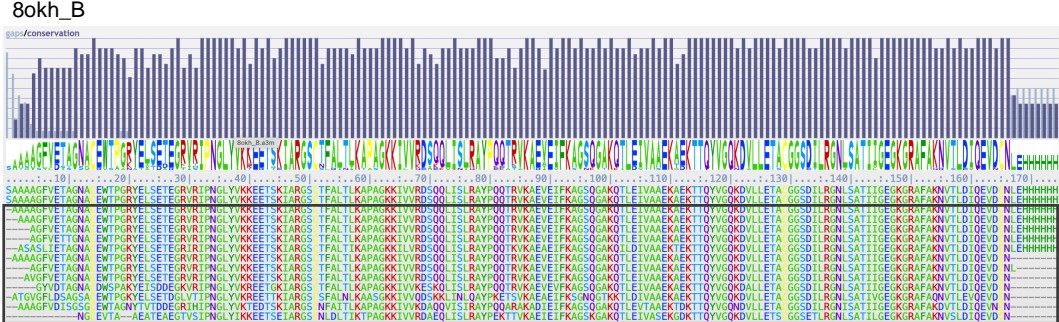

Figure 13: Augmented MSA visualization of 8okh_B.

8okw_B

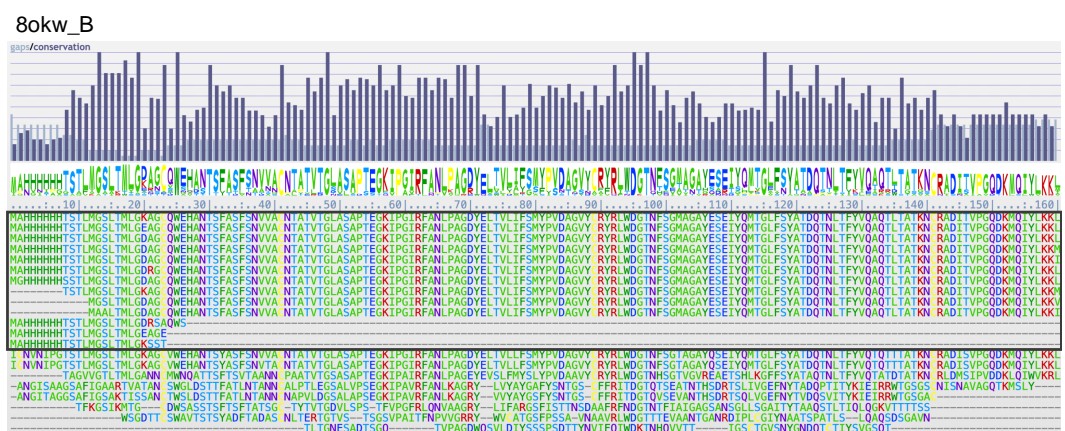

Figure 14: Augmented MSA visualization of 8okw_B.

8fih_C

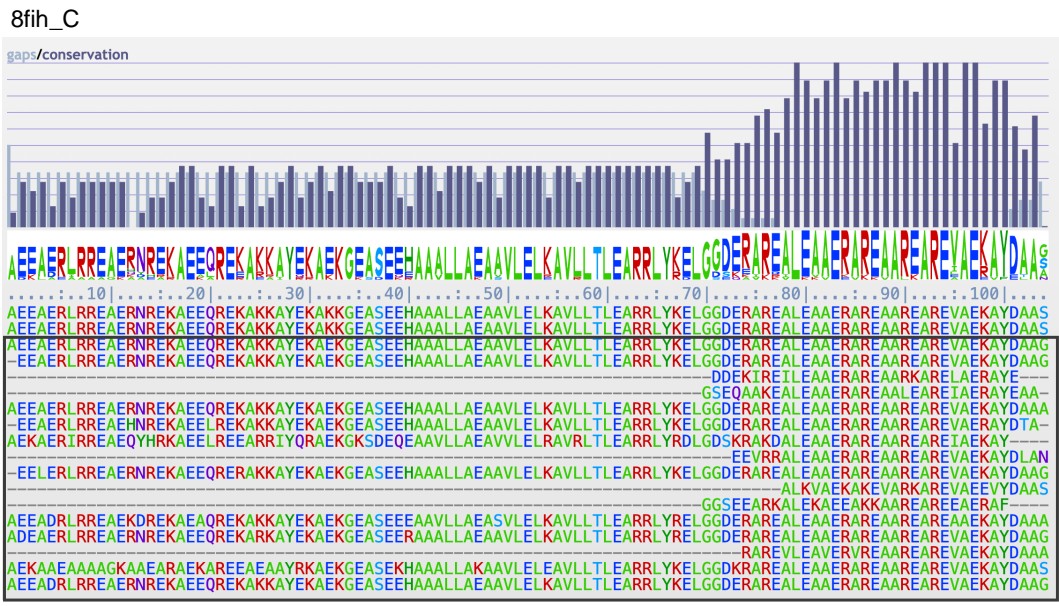

Figure 15: Augmented MSA visualization of 8fih_C.

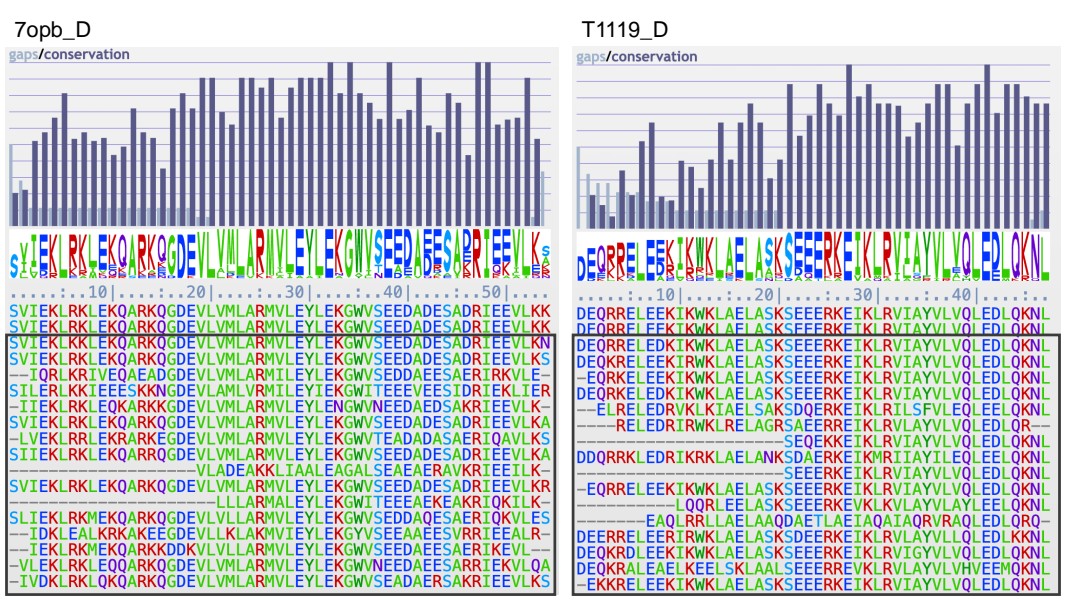

Figure 16: Augmented MSA visualization of 7opb_D and T1119_D.

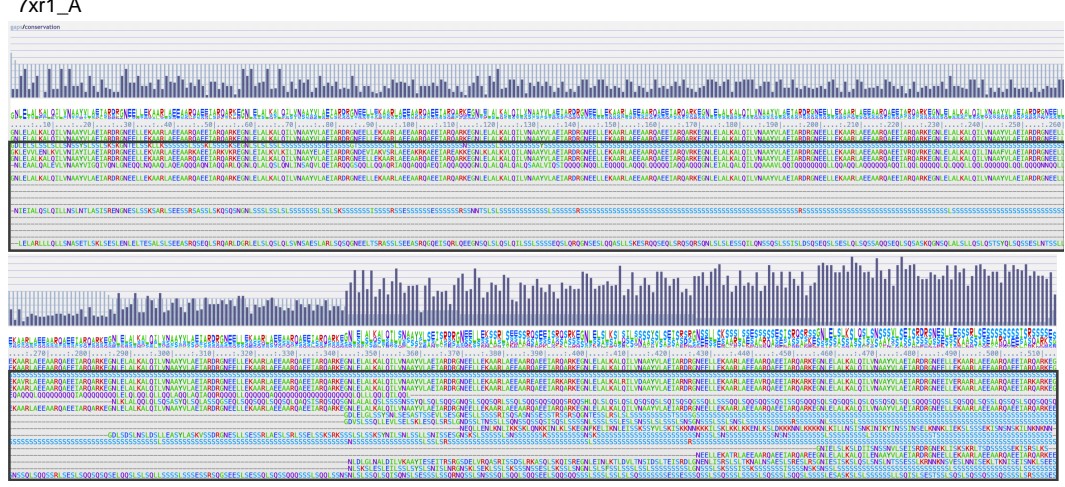

Figure 17: Augmented MSA visualization of 7xr1_A.

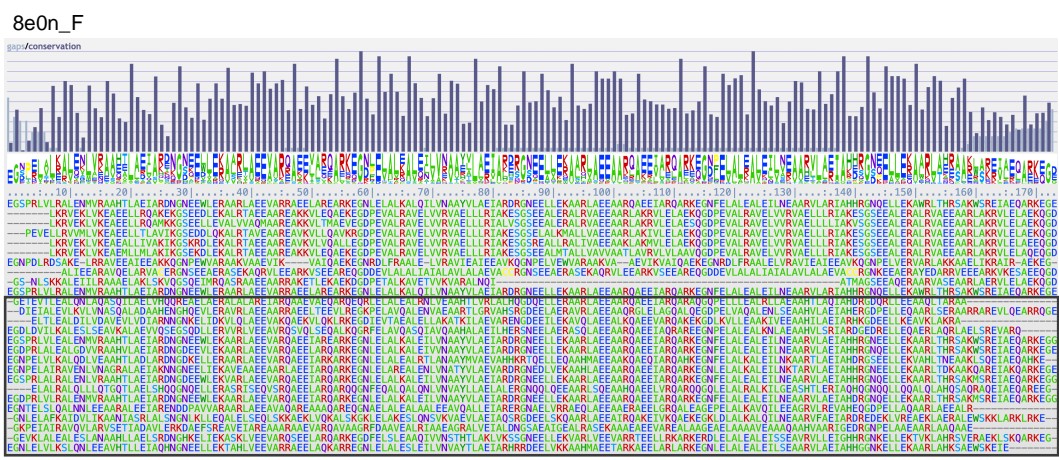

Figure 18: Augmented MSA visualization of 8e0n_F.

# I   FAILURE CASE MSA VISUALIZATION

## 7sxb_A

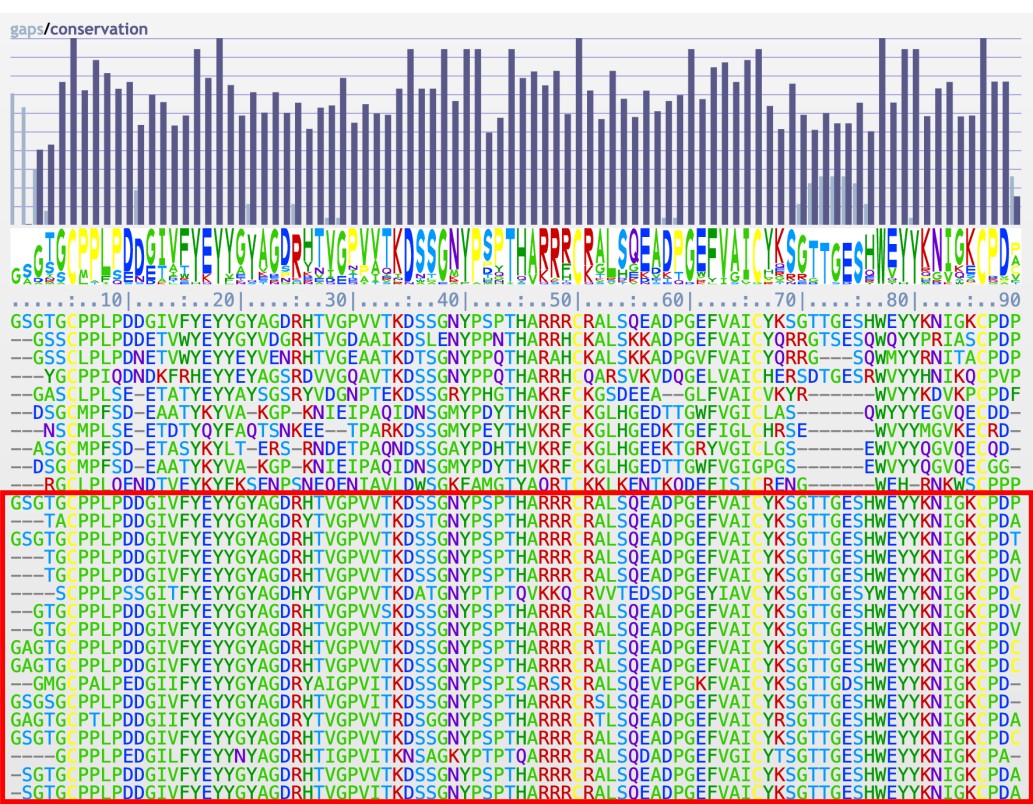

Figure 19: Failure Case MSA visualization of 7sxb_A.

## 8gzu_T3

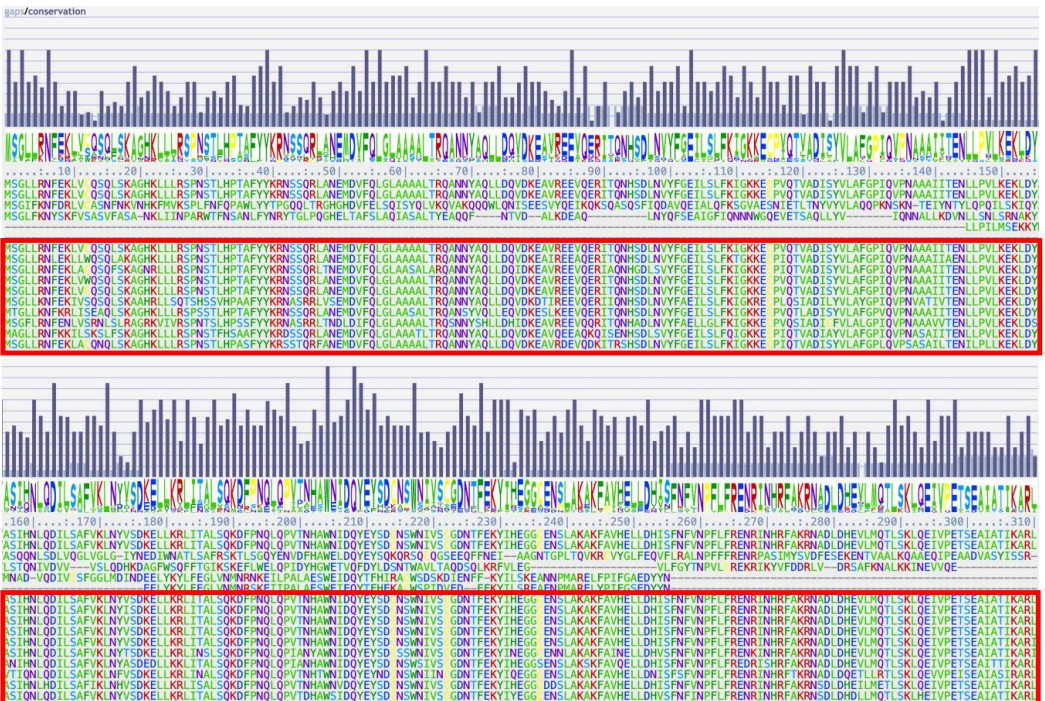

Figure 20: Failure Case MSA visualization of 8gzu_T3.

## J  USAGE OF LANGUAGE MODELS

We use large language model (LLM) to aid in the preparation of this manuscript. Its use was limited to editorial tasks, including proofreading for typographical errors, correcting grammar, and improving the clarity and readability of the text.

3bog_B

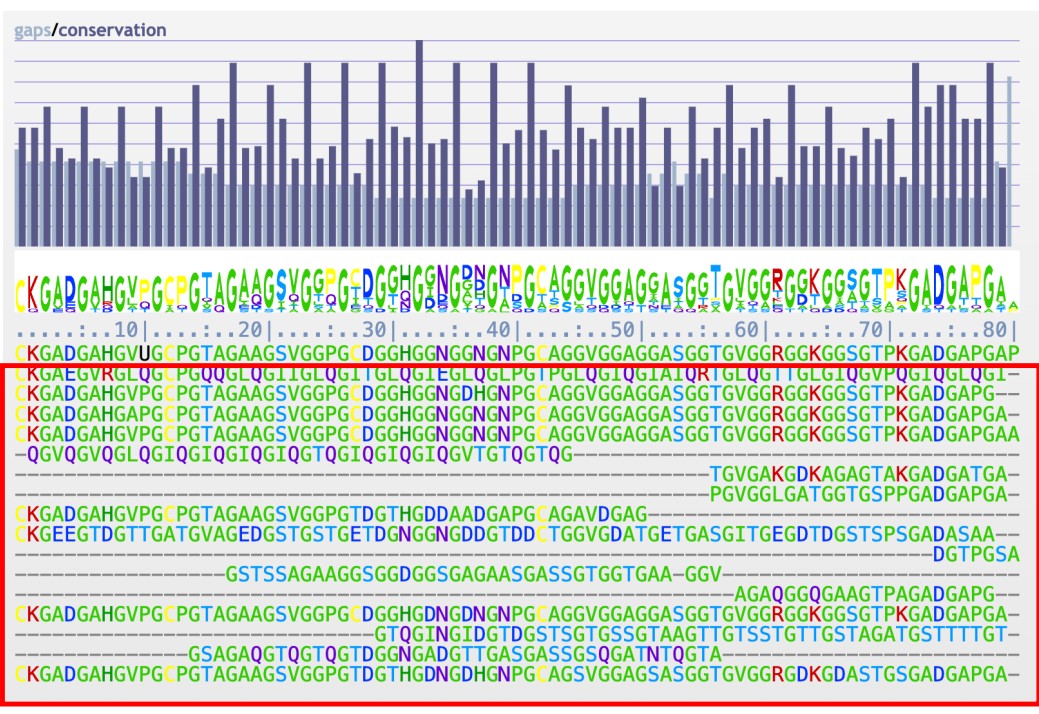

Figure 21: Failure Case MSA visualization of 3bog_B.

# 8gzu_AN

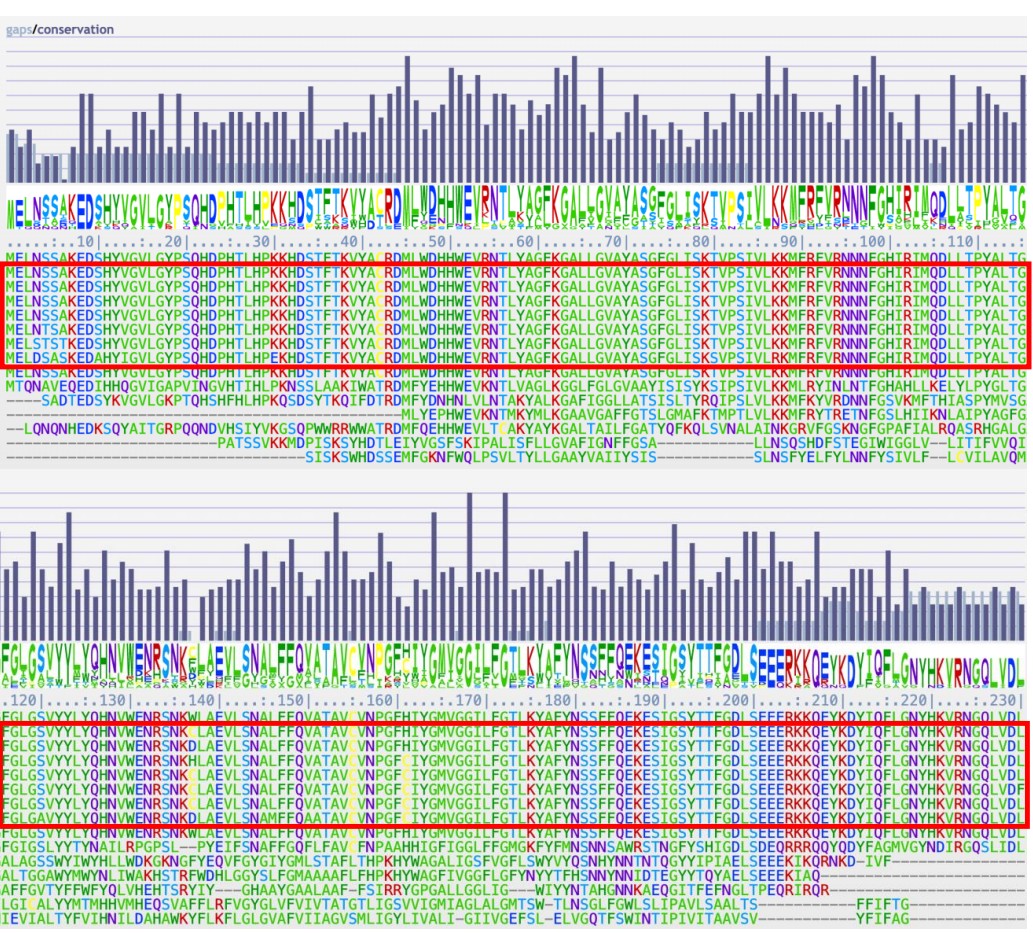

Figure 22: Failure Case MSA visualization of 8gzu_AN.

