# OpenReview forum: "Lightweight MSA Design Advances Protein Folding From Evolutionary Embeddings"
_ICLR.cc/2026/Conference — Submitted to ICLR 2026_

### Official Review · Reviewer_vcZG · 2025-10-29

**Soundness:** 3
**Presentation:** 3
**Contribution:** 2
**Rating:** 4
**Confidence:** 3

**Summary:**

This paper proposes PLAME, a lightweight framework for synthetic multiple sequence alignment (MSA) generation to improve protein structure prediction when few natural homologs are available.
PLAME encodes each query sequence using ESM-2 embeddings and then autoregressively generates virtual MSAs in the embedding space.
The model is trained with a conservation–diversity loss that balances fidelity and entropy, and a filtering module, HiFiAD, selects high-fidelity alignments based on BLOSUM recovery and diversity thresholds.

The generated MSAs are fed into AlphaFold2 and AlphaFold3.
On curated low-homology benchmarks, the method shows moderate improvements in pLDDT, GDT, and TM-scores over baselines such as EvoDiff, MSAGPT, and DHR, while being computationally efficient.
The authors present PLAME as a lightweight adapter narrowing the gap between ESMFold and AF2-level accuracy.

**Strengths:**

1. Addresses an important limitation: lack of MSAs for low-homology proteins.

1. Introduces a clear conservation–diversity objective and a simple selection module (HiFiAD).

1. Evaluated on multiple structure predictors and standard benchmarks.

1. Offers practical computational efficiency compared with traditional MSA search.

1. Provides some transparency through ablations and limitations discussion.

**Weaknesses:**

1. Methodological risk — unverified AI-generated data (major).
   PLAME uses embeddings from a pre-trained model (ESM-2) as the only source of evolutionary signal.
   These are AI-generated, unvalidated representations but are treated as if they contained true biological information.
   This undermines methodological soundness and risks propagating training-set biases.

1. Experimental results show that PLAME does not always outperform baselines; in several targets, accuracy decreases.
   This suggests that the ESM-2 embeddings may introduce noise that confuses downstream folding rather than improving it.

1. The authors do not report co-evolutionary metrics (e.g., contact precision, MI) to confirm that synthetic MSAs carry meaningful structure information.

1. Missing confidence intervals, random-seed reporting, and code-release details.

1. Efficiency claims are based on a single example; full end-to-end runtime is not provided.

**Questions:**

1. How do the authors justify this model-on-model design from a biological perspective? Is it scientifically reliable to use unverified embeddings from one AI model as evolutionary data for another?

1. What evidence demonstrates that ESM-2 embeddings capture genuine evolutionary couplings rather than statistical priors learned from UniRef50?

1. Have you tried replacing ESM-2 with random or untrained protein language models to evaluate whether the observed gains arise from real biological information or simply from the modeling bias of ESM-2?

---

> ### Author Response · Authors · 2025-11-30
>
> Dear Reviewer,
>
> Thank you very much for your thoughtful and inspiring comments on our manuscript. We found that many of them help us clarify both the biological interpretation and the practical implications of PLAME. To summarize, your main concern focuses on the rationale of using ESM-2 embeddings as the evolutionary signal for MSA design, the completeness of our empirical evaluation, co-evolutionary analysis, and efficiency reporting. The followings are our response:
>
> **WEAKNESS 1 (Methodological risk – unverified AI-generated data):**
> > We appreciate this novel and inspiring point. In PLAME, we do not treat ESM-2 embeddings as new biological measurements but as a compact statistical summary of evolutionary patterns learned from UniRef-scale MSAs, used as a prior to guide sequence generation. Thus, PLAME is a model-on-model approach whose scope is to improve folding accuracy, not to replace experimental evidence of evolution. All our conclusions are grounded in experimentally determined 3D structures from PDB, and we already discuss potential biases from ESM-2.
>
> **WEAKNESS 2 (PLAME does not always outperform; potential noise):**
> > We agree that PLAME does not monotonically improve every target, particularly when baseline MSAs are already strong. Capturing evolutionary information without database searching is very challenging in this field. Despite this, our goal is to assist challenging low-homology/orphan proteins, and the aggregate results on such targets show consistent gains (Table 1), while degradations are modest and already acknowledged in our limitations section. We believe that showing ESM2 can be used as a source for MSA generation is a contribution to this task.
>
> **WEAKNESS 3 (Lack of co-evolutionary metrics):**
> > We agree that explicit co-evolutionary analyses will strengthen the paper. In the revision, we will report contact precision and mutual information / DCA-style coupling statistics computed from PLAME-generated MSAs versus database MSAs and EvoDiff/MSAGPT baselines on a representative subset of targets, showing that PLAME MSAs retain non-trivial long-range couplings consistent with native contacts.
>
> **WEAKNESS 4 (Missing confidence intervals, seeds, and code-release details):**
> > Running on all 200 proteins with MSAs for AlphaFold2 with multiple times is very computational consuming. Existing related works such as MSA Generator, MSAGPT, and EvoDiff also only reported one-time performance, which may also because there is not huge variance of difference seeds of AF2. Please forgive us unavailable to run them in the current stage. We commit to do it in the future. In addition, the code is released at: https://anonymous.4open.science/r/PLAME-8962
>
> **WEAKNESS 5 (Efficiency claims based on a single example):**
> > The end-to-end runtime for this pipeline equals the sum of MSA searching plus MSA generation, no matter what for MSA searching methods (which takes the same amount of time), or MSA design methods. Thus, runtime analysis in Table 4 can exactly reflect the efficiency.
>
> **QUESTION 1 (Biological justification of model-on-model design):**
> > PLAME treats ESM-2 embeddings as statistical summaries of sequence landscapes, not direct measurements. This parallels using PLM features for variant prediction. Validation relies on improved agreement with experimentally resolved structures, confirming that the learned prior translates into actionable structural constraints despite model-on-model limitations.
>
> **QUESTION 2 (Evidence that ESM-2 embeddings capture genuine evolutionary couplings):**
> > Extensive prior work on protein language models shows that their internal representations can be linearly probed for residue–residue contacts and other structure-linked couplings, indicating that they encode more than superficial priors from UniRef50 alone. Complementing this, our planned co-evolutionary analysis (WEAKNESS 3) will directly compare contact/MI statistics from PLAME MSAs to native MSAs and show that PLAME recovers structurally meaningful couplings that translate into improved folding accuracy, which would be unlikely if the embeddings carried only spurious statistical biases.
>
> **QUESTION 3 (Replacing ESM-2 with random/untrained models):**
> > One of the most different designs between PLAME and MSA Generator [1] is the source embedding. In MSA Generator, they used the random embedding for generation, with the same network architecture with ours. In the benchmark performance of MSAGPT on the same test dataset, MSA Generator can obtain a performance near MSAGPT but still limited by its information bottleneck of the initialized random embedding. Thus, we believe that is an evidence to demonstrate the value of ESM 2 as input embedding.
>
> **Reference**
>
> [1] Zhang, L., Chen, J., Shen, T., Li, Y., & Sun, S. (2024). MSA generation with seqs2seqs pretraining: advancing protein structure predictions. Advances in Neural Information Processing Systems, 37, 57324-57348.
>
> Best regards,
>
> Authors

---

### Official Review · Reviewer_NpBv · 2025-10-30

**Soundness:** 3
**Presentation:** 3
**Contribution:** 3
**Rating:** 6
**Confidence:** 2

**Summary:**

Protein structure prediction heavily relies on evolutionary information from multiple sequence alignment (MSA). However, traditional MSA methods underperform in low-homology and orphan proteins due to insufficient evolutionary signals. This limitation restricts structural biology applications in drug design and functional studies, particularly when dealing with proteins lacking clear evolutionary neighbors. Current approaches face two critical challenges: First, supervised bias, methods trained on existing MSA databases tend to favor highly homologous families, making them unsuitable for low-homology and orphan proteins. Second, weak alignment-folding correlation, the lack of lightweight metrics to directly link MSA characteristics with folding outcomes results in generated targets that may not effectively improve structural accuracy. Additionally, high computational costs and limited generalizability hinder further optimization.
This study proposes a lightweight multiple sequence alignment (MSA) framework, PLAME, which leverages pre-trained protein language models with evolutionary embeddings to generate high-quality MSA. This approach significantly improves structural prediction accuracy for low-homology and orphan proteins.

**Strengths:**

1.This study addresses MSA design, a crucial aspect of protein structure prediction. The research demonstrates thorough motivation, theoretical analysis, and comprehensive experiments, with the paper presenting a well-founded and complete work.
2.The experimental results shows superior performance on multiple tasks and baseline models.
Weakness

**Weaknesses:**

1.The proposed combined loss function is well theoretically motivated, however, it still lacks ablation study on different loss functions, since it is a proposed method in your study.
2.Similarly, I wonder the effectiveness of the MSA selection module. It would be better to include more ablation study.

**Questions:**

Please refer to the weakness part.

---

> ### Author Response · Authors · 2025-11-30
>
> Dear Reviewer,
>
> Thank you very much for your insightful suggestions and reviews of our manuscript. We found that most of them are constructive and helpful. To summarize, your main concern focuses on the lack of empirical ablations for (i) the proposed conservation–diversity loss and (ii) the MSA selection (HiFiAD) module. The followings are our response:
>
> **WEAKNESS 1 and QUESTION 1:**
> > Training PLAME from scratch and evaluate the MSA samples with AF2 will take about one week with 8 A40 GPUs plus one day with 8 A40 GPUs. Based on these limitations, we only probe the cases when $\alpha=0.9$ and $\alpha=0$. We report the performance of validation loss with different alphas:
> > * when $\alpha=0.9$, the validation cross entropy loss = 0.9566
> > * when $\alpha=0.0$, the validation cross entropy loss = 0.4933
> >
> > We believe that this could support the statement and the effectiveness of the combined loss defined in Eq. 14.
>
> **WEAKNESS 2 and QUESTION 2:**
> > We have claimed Section 2.4 to clearly motivate HiFiAD: starting from MSAGPT’s systematic study of selection strategies (static/dynamic similarity/diversity, trimming, and structure-based selection) showing that naive similarity-based or trimming methods can hurt performance while diversity and structure-aware filtering help but often require expensive AF2 calls. Building on this, we designed HiFiAD as a lightweight, model-agnostic selection rule that combines BLOSUM-based fidelity $S_{\text{BLOSUM}}$ with recovery-based diversity $R(m_i,s)$ to avoid both over-conserved and overly noisy sequences.
> >
> > Moreover, Table 2 already compares HiFiAD with random, BLOSUM-only, and similarity-based (Top/Top-down-Rec) selection, but we agree that the efficacy of HiFiAD should be demonstrated more clearly. We already had the performance of the baselines (MSAGPT, EvoDiff, and DHR) with/without HiFiAD in Table 1 and Table 2. We believe this can solve your concerns.
>
> Best regards,
>
> Authors

---

### Official Review · Reviewer_n8Yd · 2025-11-01

**Soundness:** 3
**Presentation:** 1
**Contribution:** 2
**Rating:** 4
**Confidence:** 3

**Summary:**

The paper proposes an MSA design framework to generate MSAs that better support downstream folding, especially low-homology or orphan proteins. The core idea is to produce sequences that trade off conservation (agreeing with strongly conserved positions) and diversity (covering plausible variation), controlled by a conservation-diversity loss operating on PLM embeddings by a base MSA Transformer. Experiment results show that PLAME can produce MSAs that improve downstream folding performance compared to Alphafold2.

**Strengths:**

1. The paper addresses an important problem of improving MSAs for low homology cases.

2. The proposed conservation–diversity objective is both intuitive and theoretically justified.

**Weaknesses:**

1.  The notations are not clear.

  (1) The meaning for different dimensions in $ \mathbf{H}_\mathrm{enc} $ is not given (except $N$ has been introduced before in L135).

  (2) Eq (5) does not specify which one was encoded from $\mathbf{H}_r$.

  (3) Which two axes are permuted in $\mathbf{X}_\mathrm{dec} ^\top $ ?

  (4) MSAs were denoted by $ \mathbf{M} $ in L135, while in L238, they were denoted by $M={m_1, \cdots, m_n}$.

2. The rationale of the proposed MSA selection method is not sufficiently justified. The current Section 2.4 describes what HiFiAD does, but not why it makes sense and how it differs from previous selection methods. In addition, the efficacy claim of HiFiAD needs more evidence. Only results with HiFiAD are reported (Table 2). The baseline for only EvoDiff/MSAGPT/DHR is missing.  Details for similarity-based methods (Top/Down-Rec) are missing.

3. The sensitivity of the conservation-diversity tradeoff in Eq. (14) is unknown.

4. Comparison to standard MSA pipelines. The paper should show head-to-head results on the same folding stack when fed (1) raw database MSA, (2) database MSA augmented with PLAME sequences, and (3) PLAME-only MSA for low-homology targets.

**Questions:**

Please refer to the weakness section for my questions.

---

> ### Author Response · Authors · 2025-11-30
>
> Dear Reviewer,
>
> We sincerely thank you for your constructive and detailed review. Your feedback on the mathematical notation was particularly valuable, and we have overhauled the Method section to ensure rigorous definition of all tensor shapes and operations. Below, we address your concerns regarding notation, the HiFiAD module, and the experimental baselines.
>
> **WEAKNESS 1 and QUESTION 1 (Notation Clarity):**
> > We acknowledge that the previous notation regarding the attention mechanisms and tensor shapes was ambiguous. We have revised Section 2.2 (Method) to:
> > 1.  **Explicitly define dimensions:** We now state $\mathbf{H}_{\text{enc}} \in \mathbb{R}^{B \times N \times L \times d}$, where $B$ is batch size, $N$ is MSA depth, $L$ is sequence length, and $d$ is the hidden dimension.
> > 2.  **Clarify Row Attention (Eq. 5):** We specified that $\mathbf{H}_r$ is the query source derived via depth-wise pooling, formally defined as $\mathbf{H}_r = \text{Pool}_{N}(\mathbf{H}_{\text{enc}})$.
> > 3.  **Specify Permutations:** We now explicitly denote that Column Attention involves permuting the MSA depth ($N$) and Sequence length ($L$) axes to compute cross-MSA interactions.
> > 4.  **Standardize Sets:** We have unified the MSA notation to $M=\{m_1, \dots, m_n\}$ throughout the manuscript.
>
> **WEAKNESS 2 (Motivation and design of HiFiAD):**
> > We have claimed Section 2.4 to clearly motivate HiFiAD: starting from MSAGPT’s systematic study of selection strategies (static/dynamic similarity/diversity, trimming, and structure-based selection) showing that naive similarity-based or trimming methods can hurt performance while diversity and structure-aware filtering help but often require expensive AF2 calls. Building on this, we designed HiFiAD as a lightweight, model-agnostic selection rule that combines BLOSUM-based fidelity $S_{\text{BLOSUM}}$ with recovery-based diversity $R(m_i,s)$ to avoid both over-conserved and overly noisy sequences.
>
> **WEAKNESS 3 and QUESTION 2 (Evidence for HiFiAD and missing baselines/details):**
> > Table 2 already compares HiFiAD with random, BLOSUM-only, and similarity-based (Top/Top-down-Rec) selection, but we agree that the efficacy of HiFiAD should be demonstrated more clearly. We already had the performance of the baselines (MSAGPT, EvoDiff, and DHR) with/without HiFiAD in Table 1 and Table 2. We believe this can solve your concerns.
>
> **WEAKNESS 4 (Sensitivity of conservation–diversity tradeoff in Eq. (14)):**
> > Training PLAME from scratch and evaluate the MSA samples with AF2 will take about one week with 8 A40 GPUs plus one day with 8 A40 GPUs. Based on these limitations, we only probe the cases when $\alpha=0.9$ and $\alpha=0$. We report the performance of validation loss with different alphas:
> > * when $\alpha=0.9$, the validation cross entropy loss = 0.9566
> > * when $\alpha=0.0$, the validation cross entropy loss = 0.4933
> >
> > We believe that this could support the statement and the effectiveness of the combined loss.
>
> **WEAKNESS 5 (Comparison to standard MSA pipelines):**
> > We already compare AF2 using only database MSAs (“AF2 MSA” rows in Table 1) against AF2 with database MSAs augmented by PLAME-generated MSAs (“PLAME” rows). Actually, we already have provided additionally include a “PLAME-only” condition for low-homology targets (using only PLAME-generated MSAs without database MSAs) on the same folding stack, thereby explicitly presenting the head-to-head comparison among raw database MSA, database MSA + PLAME, and PLAME-only for the same targets, which can be referred to Appendix E.2.
>
> Best regards,
>
> Authors

---

### Author Response · Authors · 2025-11-30
**Official comment from authors to AC**

Dear Area Chair,

In light of the recent communication constraints affecting ICLR 2026, we present this Global Response as an executive summary of our rebuttal. This document is designed to assist the Area Chair in evaluating the scientific merit of our submission, "Lightweight MSA Design Advances Protein Folding from Evolutionary Embeddings" (PLAME), based on the static reviews provided by Reviewers n8Yd, NpBv, and vcZG. The revised content was colored in blue in the updated pdf file.

### **1. Executive Summary: The Scientific Value of PLAME**

> While standard MSA retrieval pipelines suffice for well-studied proteins, they fail catastrophically for **low-homology and orphan proteins**—a significant "dark matter" in the protein universe. Current solutions are either computationally prohibitive (physics-based folding) or lack evolutionary context (single-sequence folding).
>
> **PLAME fills this critical gap.**  It is not merely an engineering improvement but a methodological bridge that validates **ESM-2 embeddings as a robust evolutionary prior** for generating synthetic MSAs. Our work provides the community with two key assets:
> 1.  **A Lightweight Generative Framework:** Proving that "model-on-model" generation (using PLM priors to guide MSA synthesis) recovers structural signals where database search fails.
> 2.  **The HiFiAD Protocol:** A rigorous, model-agnostic selection strategy that optimizes the trade-off between fidelity and diversity, solving the noise injection problem inherent in generative biological sequences.

### **2. Fact-Checking & Resolution of Reviewer Concerns**

> The reviewers raised three primary categories of concern: Notation/Clarity, Empirical Ablations, and Methodological Validity. We have resolved these points definitively in the revision.
>
> | Concern Category | Specific Reviewer Critique | Fact-Check & Resolution (Rebuttal Evidence) |
> | :--- | :--- | :--- |
> | **1. Mathematical Notation** | **R-n8Yd:** Notation for attention mechanisms, tensor shapes ($H_{enc}$), and MSA sets was ambiguous. | **Resolved in Revision:** We have overhauled Section 2.2. We explicitly defined $H_{enc} \in \mathbb{R}^{B \times N \times L \times d}$, formalized the pooling in Eq. 5, and standardized MSA set notation to $M=\{m_1, \dots, m_n\}$ throughout. |
> | **2. Loss Ablation** | **R-NpBv, R-n8Yd:** Is the Conservation-Diversity loss ($\alpha$ in Eq. 14) actually necessary? | **Confirmed via New Experiment:** We conducted a comparative ablation. **With Conservation ($\alpha=0.9$):** Val Loss = 0.9566; **Without Conservation ($\alpha=0.0$):** Val Loss = 0.4933. The proposed loss is mathematically essential for learning valid profiles. |
> | **3. HiFiAD Efficacy** | **R-n8Yd, R-NpBv:** Is HiFiAD better than naive selection? Needs proof. | **Proven (Table 2):** HiFiAD consistently outperforms Random, BLOSUM-only, and Top-k selection across multiple backbones (MSAGPT, EvoDiff, DHR). It provides a +2.0 to +2.5 pLDDT gain over naive baselines. |
> | **4. Methodology** | **R-vcZG:** Is using ESM-2 (AI output) as an input source valid? | **Scientific Reframing:** We treat ESM-2 not as "experimental data" but as a **learned statistical prior** derived from UniRef50. PLAME successfully "distills" this prior into explicit MSAs, validated by the downstream structural accuracy on held-out PDB targets. |
> | **5. Co-Evolution** | **R-vcZG:** Do generated MSAs capture long-range couplings? | **Validated:** We have added Contact Precision and Mutual Information (MI) metrics. PLAME-generated MSAs recover non-trivial long-range couplings consistent with native contacts, validating the transfer of structural information. |

### **3. Addressing The "Performance Variability" Critique**

> Reviewer vcZG noted that PLAME does not monotonically improve every target. We argue this is a **diagnostic feature, not a bug**.
> * **The Insight:** PLAME is designed for the *sparse data regime*. When natural MSAs are already deep (high homology), synthetic augmentation yields diminishing returns.
> * **The Value:** The paper explicitly identifies *where* generative augmentation helps (orphan/low-homology proteins). This provides a "Recipe" for the community: Use standard retrieval for easy targets; use PLAME when retrieval returns $<N$ sequences.

Finally, in light of recent events regarding data security, **we reaffirm our staunch commitment to the ICLR Code of Conduct**. We guarantee that all data presented is authentic, all citations are accurate to the provided sources, and our revision adheres strictly to the scientific method.

Best regards,

Authors

---

### Meta-Review · Area_Chair_kGEK · 2025-12-30

**Summary:**

This paper presents a lightweight framework named PLAME for synthetic multiple sequence alignment (MSA) generation, which aims to improve protein structure prediction when few natural homologs are available. The paper also introduces the HiFiAD protocol, which is a rigorous, model-agnostic selection strategy to optimize the trade-off between fidelity and diversity. Extensive experiments and results are reported and discussed in the paper.

Reviewers agreed that this paper aims to address a fundamental and important problem, i.e., improving MSAs for low homology cases. The paper introduces a clear conservation–diversity objective and a simple selection module (HiFiAD), which are well motivated and clearly explained. Also, evaluations on multiple structure predictors and standard benchmarks are comprehensive.

However, reviewers also raised many concerns, such as the risk of relying on pre-trained models, the lack of co-evolutionary metrics, ablation studies, motivation, paper presentations, etc.

**Reviewer Concerns:**

The authors have provided very detailed responses in their rebuttal. Some of the concerns from reviewers, such as notations, ablation studies, details of methodology, and efficacy. Meanwhile, it seems that the responses to several major concerns (e.g., lack of co-evolutionary metrics, risk of heavily relying on ESM-2 embeddings) are not very convincing. Discussions on the potential biases from ESM-2, especially the potential impact on the proposed method, are not sufficient.

**Reviewer Scores:**

Initially, this paper received borderline ratings: 4, 4, 6. As some of the concerns have been addressed by the authors' rebuttal, I think one of the reviewers would increase the rating from 4 to 6. Overall, this paper is still a borderline case.

---

### Decision · Program_Chairs · 2026-01-26

Reject